# Concurrent measurement of working memory and inhibitory control and their correlations with autistic and ADHD traits in the general population

**Yasamin Rahmati***, **Christopher Jarrold**

School of Psychological Science, University of Bristol, Bristol, United Kingdom

* JV22264@bristol.ac.uk

## Abstract

Executive function can be defined as the combination of goal representation in working memory and the inhibition of goal-irrelevant responses. This paper comprises two complementary studies that assess these core components of executive function orthogonally and examine their correlation with ASD and ADHD traits in the general population. Both studies utilized a paradigm with two tasks, each assessing working memory and one type of inhibitory control concurrently: the modified flanker task, which measured working memory and interference control, and the modified spatial conflict task, which measured working memory and response inhibition. The aim was to explore the main effects of memory and inhibitory load in each task, investigate potential over-additive interactions between executive function components, and examine the correlations between autistic and ADHD traits and task performance. Each study involved 100 neurotypical adult participants. In Study 1, results showed that reaction time in the flanker task was significantly influenced by demands on both working memory and inhibitory control, whereas in the spatial conflict task only the inhibitory manipulation produced the expected effect. Study 2 introduced modifications that revealed effects on reaction time in the spatial conflict task due to both memory and congruency load. The flanker task demonstrated memory effects in reaction time, but congruency effects were only evident under low memory conditions. No interactions between executive function components in an over-additive way were observed in either Study 1 or Study 2. Bayesian linear regression and correlation analyses found evidence against any meaningful correlations between the size of the congruency or memory effect, computed for any dependent variable and ASC or ADHD traits in both studies.

## Introduction

Executive function (EF) is an ability that helps monitor and regulate goal-directed behaviour to adapt to environmental demands [1], especially in situations in which

**Data availability statement:** Data availability The data for Study 1 are available at: https://osf.io/2eh7m/files/osfstorage/69221e00700e98148d11749c The data for Study 2 are available at: https://osf.io/5cqha/files/osfstorage/69221f0d6bfdf0bd19e640d4.

**Funding:** The author(s) received no specific funding for this work.

**Competing interests:** The authors have declared that no competing interests exist.

people need to concentrate or avoid relying automatic responses [2–4]. It is an umbrella term that covers various cognitive processes, including, working memory (WM), inhibitory control (IC), and cognitive flexibility [5] with some accounts also including the related concepts of planning, attention, self-monitoring, self-regulation, and initiation [6,7]. The current study focuses on working memory and inhibitory control, which are regarded as core components of executive function in a number of influential theoretical frameworks [8–10]. Working memory involves the ability to maintain and manipulate online information [11] and inhibitory control is the ability to cancel or suppress an action that is irrelevant, no longer needed, and/or inappropriate. Although various classifications have been proposed to categorize inhibitory control [12–14], there is substantial evidence for it being a multidimensional construct including both response inhibition, which involves the ability to suppress automatic but inappropriate responses, and interference control, which is the ability to filter out irrelevant but conflicting information [12,15–17].

Executive function skills are essential for mental and physical well-being, educational achievement, and overall success in life, as well as for social, psychological, and cognitive development [8,18]. More specifically, executive functions play a crucial role in developing and executing plans, forming analogies, adhering to social rules, solving problems, adapting to unexpected situations, multitasking, and retrieving memories effectively [19]. Understanding executive functions is therefore important, particularly for individuals who experience difficulties in these skills, including those in neurodiverse groups. Extensive research shows that neurodivergent individuals often perform differently from typically developing individuals on tasks assessing various aspects of executive function [20–24]. Executive function strengths and difficulties have traditionally been studied within a categorial deficit framework that sought to identify specific executive function 'impairments' associated with specific neurodevelopmental conditions (NDCs) [25,26]. Within this perspective, research has consistently highlighted that challenges in executive function are frequently associated with two prevalent neurodevelopmental conditions: autism spectrum condition (ASC) and attention-deficit/hyperactivity disorder (ADHD) [21,23,27]. The following sections review executive function findings in ASC and ADHD within this traditional diagnostic view, before turning to more recent transdiagnostic perspectives that move beyond these boundaries.

For autism, early investigations of executive function primarily emphasized cognitive inflexibility, particularly challenges with set-shifting and adapting to new information [28,29]. However, despite substantial research (including several meta-analyses and systematic reviews) evidence for executive dysfunction in autism remains mixed. For instance, while some studies, such as the meta-analysis by Demetriou et al. [25], have reported moderate executive function difficulties across all subdomains in autism compared to neurotypical controls, others have suggested challenges only in specific domains. For instance, Guerts [30] reported that children with a diagnosis of high-functioning autism (HFA) exhibited compromised performance across all major executive function domains except for interference control and working memory.

Similarly, the literature on executive dysfunction in ADHD presents a complex picture. Although the condition is classically defined by impulsivity and inhibitory 'deficits' [31,32], evidence for other components of executive function is mixed. A meta-analysis of experimental executive function measures reported significant difficulties in response inhibition, vigilance, working memory, and planning, while findings on cognitive flexibility were less consistent [32]. Cognitive flexibility, in particular, shows the greatest variability, with some studies finding no significant difficulties [33], while others describe reduced performance in this area [34].

Findings from studies focusing directly on working memory and inhibitory control (which are the focus of this paper) also show inconsistent results for both ASC and ADHD. In autism, some studies have reported difficulties in prepotent response inhibition and interference control [35,36], while others have not [37,38]. More recently, a meta-analysis by Tonizzi et al. [39] identified challenges in both aspects of inhibitory control among individuals with an autism diagnosis. Similarly, findings on working memory are inconsistent. While some studies have suggested intact working memory in individuals with an autism diagnosis [40], others have reported relative difficulties, particularly in spatial working memory [37,41]. A meta-analysis by Habib et al. [42] further reported significant challenges in both phonological and visuospatial working memory in individuals with an autism diagnosis compared to controls.

In ADHD, although inhibitory control difficulties are widely acknowledged [43–45] findings still vary considerably across studies. For instance, Elosúa et al. [46] reported relative difficulties in working memory, attentional shifting, and updating but not in inhibition. Similarly, evidence for working memory function is mixed. While many studies report difficulties in working memory for both children and adults with ADHD [47,48], not all studies support this view. For example, Superbia-Guimarães et al. [49] found that participants with ADHD performed comparably to neurotypical peers on verbal working memory tasks, particularly when the task structure facilitated 'attentional refreshing' as a means of maintaining to-be-remembered information.

The mixed patterns observed in prior executive function research in neurodevelopmental conditions may be due to several factors that make studying these processes in these populations challenging. One contributing factor is the fundamental heterogeneity inherent to neurodiverse populations [50]. Individual differences contribute substantially to executive function variability, and both autism and ADHD show substantial within-group heterogeneity [51]. As both conditions are increasingly recognised as spectrum conditions, individuals can present with a wide range of trait profiles, making the study of executive function s in these groups more complex. Notably, these spectrums extend beyond those with a formal diagnosis. In autism, for example, autistic traits appear to be continuously distributed across the general population with some individuals (particularly first-degree relatives of those with a diagnosis) exhibiting subclinical or broader autism phenotype features to varying degrees [52–54], including subtle differences in social communication and restricted, repetitive behaviours [52,53]. This has prompted researchers to examine whether executive function differences also align with this broader continuum. However, findings remain inconsistent. While some studies of first-degree relatives report intact executive performance [55–58], others suggest atypical executive function performance in these groups [54,59–62]. Similarly, ADHD traits, such as inattention and hyperactivity-impulsivity, exist on a continuum in the general population [63,64]. Research suggests that even individuals without an ADHD diagnosis but with higher ADHD traits might experience greater executive function challenges. For instance, Moses et al. [65] found that individuals with higher ADHD traits demonstrated relative difficulties in working memory and prepotent response inhibition. This association is further supported by Crosbie et al. [43], who reported that higher ADHD trait scores were associated with more significant challenges in prepotent response inhibition. Taken together, these studies suggest that executive function patterns in these conditions are not categorically present or absent but can vary across individuals with different trait profiles.

Furthermore, the substantial variability observed in executive function in individuals with autism and ADHD is not solely attributable to core neurodivergent traits but is also strongly shaped by individual contextual factors. More specifically, context shapes executive function through environmental influences, learned strategies, motivation, beliefs, and prior experiences, affecting how individuals apply executive skills in different situations [66]. This means that executive function

variability in autism and ADHD likely reflects not only differences between individuals, but also differences in the situations in which they use these skills, further complicating the overall picture.

In addition to these individual and contextual differences, cross-condition phenotypic overlap further complicates studying executive function in neurodiverse populations. Autism and ADHD share significant clinical and neurobiological similarities, and many individuals present traits associated with both conditions [51]. Furthermore, many individuals exhibit overlapping traits from other conditions, which can compound executive function challenges [67]. The broad range of traits and their co-occurrence across conditions strongly suggest that executive function difficulties represent a transdiagnostic feature of neurodevelopmental conditions rather than specific difficulties tied to a particular single diagnosis. Consistent with this, a systematic review and meta-analysis by Sadozai et al. [67] reported a moderate executive function delay across all neurodevelopmental conditions (g = 0.56, 95% CI [0.49–0.63]) compared to controls, with even larger effects observed when co-occurring conditions were present were present (g = 0.72, 95% CI [0.59–0.86]).

In response to this transdiagnostic perspective, research has increasingly shifted toward approaches that move beyond traditional categorical comparisons [68]. The transdiagnostic framework leverages the continuous nature of trait expression to explore shared characteristics and underlying mechanisms across multiple neurodevelopmental conditions, rather than contrasting diagnostic categories with neurotypical control groups. It acknowledges that executive function strengths and difficulties are not confined to specific conditions but are dimensionally distributed across the population, varying in degree rather than being determined by formal diagnosis. Recognising the interconnected relationship between autism spectrum conditions and ADHD is therefore crucial within this emerging framework. Accordingly, the current study adopts a dimensional approach, examining ASC and ADHD traits along a continuum within the general population to explore their associations with key executive function components, rather than using a categorical approach that compares ASC or ADHD to control groups.

Beyond heterogeneity, contextual influences, and cross-condition overlap, another major factor contributing to the mixed and inconsistent findings in executive function research is the lack of a clear consensus on how executive function should be defined and conceptualised [50,69]. Such definitional variation leads to differences in study outcomes, as researchers measure executive function components according to the particular theoretical framework they adopt [50,25]. Some research on executive function among developmental conditions assesses strengths and weaknesses at the whole-system level [70], whereas other work focuses on specific skills such as inhibition [31] or working memory and inhibition [23,71]. This reflects differences between theoretical models, with some proposing a unitary system, while others argue for distinct, fractionable executive functions such as working memory and inhibitory control. The unitary models propose that working memory and (either type of) inhibitory control rely on a shared, limited-capacity system [72,73]. A direct consequence of this competition for shared resources is that increasing inhibitory or working memory demands should lead to a greater susceptibility to the effects of the other factor [74,75]. In other words, an (over-additive) interaction is predicted, with the effects of each factor becoming more pronounced as their combined demands increase and deplete the shared resource. As a result, under a unitary model of executive function the combined impact of two or more executive demands would be greater than expected from adding their individual effects alone.

In contrast, fractionated models suggest that executive function components (specifically working memory and the two forms of inhibitory control in this study) are distinct executive functions [76]. Evidence for fractionated models comes from factor-analytic studies across age groups [66, 77–80]. According to these fractionated models, increasing demands on working memory or on either form of inhibitory control affect performance independently, implying additive effects, that is, the combined impact of both demands equals the sum of their separate effects. Preliminary evidence, in line with a fractionated model, suggests that such interactions between executive function components may not always emerge. For instance, Jarrold et al. [9] used a single task to examine working memory and response inhibition in children and found that these components operated independently rather than amplifying one another's effects. However, this evidence is limited: the study focused solely on children and examined only the relationship between working memory

and response inhibition. This highlights the need to investigate these associations in adults as well, and to assess both forms of inhibition (response inhibition and interference control) using appropriate, novel tasks designed to target each component.

To address the above limitations in the measurement and conceptualisation of executive function, the present research employed a novel approach that provides a more integrated assessment of executive function and, therefore, contributes to a deeper understanding of this construct. This paper comprises two complementary studies, each using a single-task structure to assess working memory and inhibitory control simultaneously, rather than measuring them separately with distinct tasks (see Jarrold et al. [9]). Because two forms of inhibitory control were of interest, each study employed two versions of the central task: one combining working memory with response inhibition, the other combining working memory with interference control. Assessing these processes within the same task structure allows their potential interactions to be examined directly, contributing to ongoing debates about the structure of executive function. It also overcomes key limitations of traditional approaches that rely on separate tasks to measure each component. More specifically, utilizing distinct tasks is not only time-consuming [81], but also introduces confounding effects as each task will have unique characteristics or demands that could influence performance, making it challenging to isolate the specific contributions of each core function. Additionally, traditional tasks may not be process-pure; for example, memory-based tasks can involve inhibitory aspects, and inhibitory tasks may require memory for rules [82].

Furthermore, in line with the transdiagnostic perspective introduced above, the current work also investigated associations between executive function performance and extended ASC and ADHD traits in the general population. Study 1 examined ASC traits exclusively, using a dimensional rather than case–control approach, consistent with a transdiagnostic framework. Given the substantial co-occurrence of ASC and ADHD and their shared executive function characteristics, Study 2 expanded the scope to include both ASC and ADHD traits, thereby providing a more robust transdiagnostic perspective.

To summarise, this paper aimed to assess working memory and two types of inhibitory control concurrently within a single paradigm and to investigate possible over-additive interactions between these processes ( [72], though see Jarrold et al. [9]). A further aim was to explore associations between task performance and ASC and ADHD traits in the general population.

The preregistered hypotheses predicted that working memory and both forms of inhibitory control (response inhibition and interference control) would show robust effects when measured together in a single task. Specifically, slower reaction times (and higher inverse-efficiency scores) and lower accuracy were expected in high-memory compared to low-memory conditions, and in incongruent compared to congruent trials.

Regarding their interaction, and drawing on findings from Jarrold et al. [9], it was anticipated that working memory and (either types of) inhibitory control would operate independently rather than interact. The present research extends Jarrold et al.'s approach by examining two forms of inhibition (response inhibition and interference control), testing adults rather than children, and employing novel flanker and spatial-conflict tasks.

In more practical terms, and with respect to the interaction between executive functions, it was hypothesized that the performance cost (i.e., slower reaction times/inverse efficiency and reduced accuracy) observed under high working-memory load would be similar across both congruent and incongruent conditions. Likewise, the cost of inhibition (whether response inhibition or interference control) was expected to remain consistent regardless of memory load. In other words, high demands on both working memory and either type of inhibition were not expected to produce an additional, over-additive decline in performance.

Finally, it was hypothesised that higher levels of ASC and ADHD traits would be associated with larger effects of memory load and inhibitory load, such that individuals with higher trait levels would show greater slowing of reaction times/inverse-efficiency scores and larger reductions in accuracy as task demands increased.

## Study 1

### Method

The Study 1 was preregistered on the Open Science Framework (OSF). Access to the preregistration documents for this study is available through the following link:

Study 1: https://doi.org/10.17605/OSF.IO/WA34M

This study received ethical approval from the School of Psychological Science Human Research Ethics Committee at the University of Bristol under the approval code 13573.

### Participants

In line with the preregistration, a total of 100 participants aged 18–25 were recruited from the general population through the host institution's course credit scheme at the University of Bristol. All participants were undergraduate psychology students. This sample size was chosen to provide adequate power to detect correlations of.32 or higher. After applying the relevant exclusion criteria (which differed across analyses; see relevant sections below), the final sample size varied for each analysis, and the exact numbers are reported in the corresponding sections. Participants were recruited between 7/02/2023 and 14/02/2023, with written informed consent obtained from all participants prior to the start of the study.

### Measures

**Cognitive tasks (the executive function battery) and their conditions.** The executive function battery was a novel assessment that was presented online through the Gorilla Program. This battery consisted of two tasks each of which orthogonally manipulated working memory and one type of inhibitory control. More specifically, a modified version of a spatial conflict task was used to manipulate working memory load and response inhibition load and a modified version of a flanker task was used to manipulate working memory load and interference control load. These modified tasks are respectively based on widely used congruency tasks in executive function research, namely the Simon task [83] and the Eriksen Flanker task [84].

The Simon task [83], or spatial conflict task, is consistently classified as a measure of prepotent response inhibition [85]. This refers to the ability to suppress a dominant motor response [13,86], which requires suppression of competing responses [87]. The Simon task rests on the assumption that the spatial location of the stimulus automatically activates the corresponding response option. For example, a stimulus presented on the left side of the screen automatically activates a left-hand response option. Participants are typically slower and less accurate on trials where the stimulus is presented on the opposite side to the correct response option (incongruent trials) compared to when the stimulus is presented on the same side as the correct response option (congruent trials) (see [83]). In this task, the primary measure of interest is the difference in accuracy and/or speed (reaction time) between these experimental conditions [9,88].

The Erikson Flanker task [84] is one of the most commonly used measures of interference control, which refers to the ability to ignore irrelevant information while processing target stimuli [88]. In this task, participants must respond to a target stimulus as quickly as possible while ignoring flanking distractors that either prompt the same response as the target (congruent condition) or provoke an opposite response (incongruent condition). Interference control differs from response inhibition in that it involves conflicting information that is separate from the stimulus, whereas response inhibition deals with conflicting information that forms part of the stimulus itself. The main dependent measures in this task are the differences in accuracy and reaction time between these conditions [88].

As previously mentioned, the current research introduced an additional working memory load component into the traditional versions of the Simon and Erikson Flanker tasks. Both modified tasks followed a similar structure, featuring two comparable memory load conditions and the inclusion of distractors in the spatial conflict task to maximise the superficial similarity of the two task displays and make them much more comparable. In each of the modified tasks, two memory

conditions (low memory and high memory) were included and within each memory condition there were two trial types – congruent and incongruent– that reflected the inhibitory load manipulation (whether of response inhibition in the spatial conflict task or interference control in the flanker task).

The working memory load manipulation employed in both tasks involved varying the complexity of the response rules that had to be maintained. Both memory conditions of each task required participants to remember the response associated with each of four stimuli that could appear in that condition; in each condition two stimuli were associated with one key press and the other two stimuli were associated with a different key press. In the low memory conditions of each task, these four stimulus-response mappings could readily be grouped into two rules (e.g., press W on the keyboard when you see a light or dark blue airplane and press P on the keyboard when you see a light or dark red airplane). In the high memory load conditions, participants were required to remember four different rules that could not be grouped together (e.g., press W if you see a purple or brown airplane, and press P if you see a yellow or green airplane).

All of the participants started the experiment with the flanker task (as the two tasks were quite similar in their structure but the organization of the spatial conflict task was slightly more complicated). The order of presentation of memory load conditions was counterbalanced within each task (leading to four orders of presentation of the battery). The flanker task contained 212 trials consisting of 20 practice trials and 96 trials in each of the low and high memory load conditions. The spatial conflict task contained 200 trials including 24 practice trials and 88 trials in each of the low memory and high memory load conditions. There were an equal number of congruent and incongruent trials in each memory load condition of each task.

The general structure of the study was as follows: In each trial of each task, participants first viewed two letters (W and P), with P positioned on the right and W on the left edges of the bottom of a white blank screen, presented for 500 ms. Following this, stimuli were presented. These stimuli consisted of 5 airplanes (four grey and one coloured) presented in a horizontal line at just above the midline of the screen, in such a way that the W was located under the left-most airplane and the P was located under the right-most airplane (see Fig 1). In the flanker task the airplanes were either all pointed to the right or all pointed to the left with the coloured target stimulus in the central position. During congruent trials, all stimuli pointed in the direction aligned with the correct response prompted by the central target stimulus (e.g., all pointing to the left when the correct response was to press the W key). Conversely, during incongruent trials, all stimuli pointed in the opposite direction to the correct key response associated with the central target stimulus (e.g., pointing right for a W key press). In the spatial conflict task, all the airplanes were presented in an upright position with the coloured target stimulus being presented at either the extreme left or the extreme right of the row. On congruent trials the target stimulus appeared

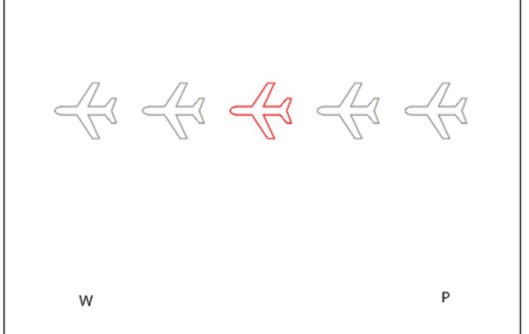

A sample trial in the Flanker task

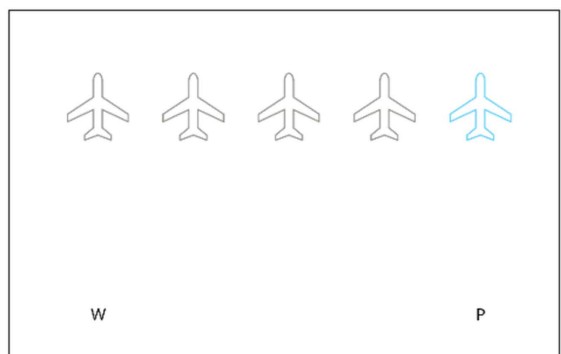

A sample trial in the Spatial Conflict task

**Fig 1. Sample trials in cognitive tasks: Screenshots from the Gorilla program.**

above the letter and response key associated with it (i.e., on the left for a W key press or on the right for a P key press); on incongruent trials the target stimulus appeared on the opposite side to its associated response (i.e., on the right for a W key press or on the left for a P key press).

Participants started each task with the aforementioned number of practice trials and they then proceeded to the either low or high memory load condition of that task. They were instructed to respond to the coloured airplane (the target stimulus) while ignoring the grey airplanes, and to make their responses as quickly and accurately as possible by pressing one of two response keys (W or P). The next trial began only after a response was made. During the practice trials of both tasks, participants received accuracy feedback for both their correct and incorrect responses, and if they made a mistake, the trial was repeated. However, in the experimental trials, they only received feedback following incorrect responses (without repeating the trial).

The same colours were used for practice trials in both tasks. Specifically, pink corresponded to P and orange to W. However, different colours were used for the target stimuli in the experimental trials. In the flanker task's low memory trials, the designated colours were light and dark blue for W (pointing either to the right or left) and light and dark red for P (also pointing either to the right or left). In low memory trials of the spatial conflict task, the chosen colours were light and dark brown for W and light and dark green for P. Given the limitation of available distinguishable colours for the high memory conditions, some colours from the low memory conditions were reused. However, great care was taken to maintain the same spatial position as in the low memory conditions. As a result, the high memory flanker task employed brown and purple (pointing either to the right or left) for W and yellow and green (pointing either to the right or left) were assigned to P. For the high-memory spatial conflict task, W was mapped to blue and purple, and P to red and yellow (it could be argued that these colours could be categorized as warm versus cool colours, which may have reduced the intended working memory demand. This potential limitation was addressed in Study 2.) Completing the battery took approximately 20–25 minutes.

**Autism traits measure (AQ-S).** In line with our preregistration, autistic traits were measured using the Short Autism-Spectrum Quotient (AQ-S [89]). The AQ-S is a shortened version of the full AQ [90], designed for convenient use in large-scale studies where the full AQ's length may be impractical. Hoekstra et al. [89] established the AQ-S structure and content through item selection and validation analyses involving individuals with ASC and control groups. The AQ was designed to capture the core dimensions of autistic traits in adults with typical levels of intelligence. The AQ-S includes 28 of the 50 original AQ items and retains its broad dimensionality. These items measure a higher-order 'Social Behaviour' factor, focusing on social skills, imagination, routine and switching, as well as a 'Numbers and Patterns' factor, which focuses specifically on an interest in numbers, dates, patterns and categories of things. Items were rated on a 4-point Likert scale (1 = definitely disagree to 4 = definitely agree). In this study, the total AQ-S score served as the primary dependent variable, calculated by summing all 28 items (range: 28–112), with higher scores indicating greater levels of autistic traits. Although only the total score was preregistered for analyses, exploratory analyses were later conducted on the two subscale scores in response to reviewer request. In ASC and control samples, CFA analyses have generally found reasonable fit of this structure for the AQ-S [89, 91].

### Analysis plan

**Primary task analyses.** For each task, Bayesian repeated-measures ANOVAs were conducted to assess the effects of memory load, congruency, and their interaction. The full set of models that can be constructed from these factors was automatically tested. This included a null model (with only subject and random slopes), models containing each factor on its own, a model containing both factors without an interaction, and a model including the interaction. These models were compared using Bayes factors ($BF_{10}$) relative to the null model, and evidence for each effect was summarised using inclusion Bayes factors (BF(incl)).

Analyses were carried out separately for three dependent variables. The main text focuses on reaction-time (RT) analyses; RT data were log-transformed to reduce positive skew (this transformation was not preregistered). Accuracy and

inverse-efficiency (IE) measures were also analysed, and full results for these variables are provided in Supplementary Materials S1 and S2 Appendices, respectively. IE scores were calculated to assess potential speed–accuracy trade-offs and were log-transformed; this addition was made in response to reviewer feedback (and was not preregistered).

**Association with ASC traits.** To examine relationships between task performance (RT, accuracy, and IE) and autistic traits (AQ-S), two types of analyses were conducted. Zero-order associations were assessed using Bayesian correlations, and partial-correlation equivalents were obtained through Bayesian linear regressions extracting memory-load and inhibition-load effects. In response to reviewer requests, both analyses were additionally repeated for the two AQ-S subscales, and the regression models were further extended to include age and gender as covariates. These supplementary analyses are reported in S4 and S5 Appendices, respectively.

**Assumptions and model diagnostics.** Assumptions and model diagnostics were examined for all analyses. For the Bayesian repeated-measures ANOVAs, independence of observations was evaluated using the Durbin-Watson test and residual plots, and sphericity was automatically satisfied as all within-subject factors had only two levels. For the Bayesian correlation and linear regression analyses, the assumptions of linearity, independence, and normality of residuals were verified. For the linear regressions, homoscedasticity and the absence of multicollinearity were also checked. No serious violations of these assumptions were detected.

**Software and priors.** All analyses were conducted in JASP version 0.17.3 [92] using the default Cauchy priors on standardized effect sizes ($r = 0.5$ for fixed effects, $r = 1$ for random effects, $r = 0.354$ for covariates). Bayes Factors for inclusion and exclusion of effects were computed using the "across matched models" option, following Keysers et al. [93]. In line with Jeffreys [94], BFs between 1 and 3 were considered weak evidence, 3–10 moderate, and >10 strong.

## Outliers and exclusions

Trial-level reaction time outliers were identified using the Median Absolute Deviation (MAD) method outlined by Leys et al. [95], applying a ± 3 MAD criterion, and the corresponding trials were removed from all analyses. This procedure was performed independently for each participant and for each trial type (congruent/incongruent) within each condition, resulting in eight MAD-based exclusions per participant. Additionally, to account for participant-level outliers, Mahalanobis distance analyses were conducted across the four RT trial types (2 memory × 2 inhibition). This final outlier exclusion step was not preregistered and was added in response to reviewer feedback to address potential concerns regarding RT variability). Participants whose values exceeded the critical threshold for $p = 0.001$ ($\chi^2 > 18.47$, df = 4) were excluded. Based on this criterion, three participants were excluded from the RT analysis of the flanker task and three participants were excluded from the spatial conflict task analysis in Study 1.

Other exclusion criteria involved the removal of individuals who exhibited less than 60% accuracy in the congruent trials of the low working memory load condition from data analysis for that specific task. Based on this criterion no individuals were excluded from either task, as everyone met this requirement. Additionally, immediate repetitions of a stimulus across consecutive trials were not intentionally avoided. However, data from such trials were omitted and were not analyzed, in accordance with Bertelson [96]. These trimming and exclusion plans were consistent with the pre-registered plan for this study.

Furthermore, although not preregistered, participants were screened for colour-blindness in pre-study questions with the intention of excluding affected individuals. No participants in Study 1 were excluded on this basis.

## Results

### Reaction time in the flanker task

Descriptive statistics for participants' reaction time in the flanker task are shown in Table 1.

A Bayesian repeated-measures ANOVA on log-transformed reaction times showed that the best-fitting model included only the main effects of memory load and congruency ($BF_{10} = 3.32 \times 10^{30}$ relative to the null model). There was overwhelming evidence for including the main effect of memory load ($BF(incl) = 2.71 \times 10^{21}$), with participants responding more quickly

**Table 1. Descriptive statistics of reaction time (RT, in milliseconds) in the flanker task (N = 96).**

| Condition | Mean | Std. deviation | Minimum | Maximum |
|---|---|---|---|---|
| **Low memory and congruent** | 481 | 69 | 345 | 772 |
| **Low memory and incongruent** | 498 | 70 | 353 | 722 |
| **High memory and congruent** | 572 | 86 | 370 | 794 |
| **High memory and incongruent** | 605 | 105 | 400 | 920 |

in low-memory trials (M = 489 ms, SD = 67 ms) than in high-memory trials (M = 589 ms, SD = 93 ms). There was also very strong evidence for including the main effect of congruency (BF(incl) = $1.11 \times 10^9$), reflecting faster responses on congruent trials (M = 524 ms, SD = 69 ms) compared to incongruent trials (M = 552 ms, SD = 77 ms). In contrast, there was anecdotal evidence *against* including the interaction between memory load and congruency (BF(excl) = 1.530). The corresponding reaction time distributions across low and high memory conditions, and for congruent and incongruent trials within each, are shown in Fig 2.

### Reaction time in the spatial conflict task

Table 2 displays descriptive statistics for participants' reaction time in the spatial conflict task. A Bayesian repeated-measures ANOVA on log-transformed reaction times showed that the best-fitting model for the spatial conflict task included the main effects of memory load and congruency, as well as their interaction (BF$_{10}$ = $3.15 \times 10^{15}$ relative to the null model). There was weak evidence for including the main effect of memory load (BF(incl) = 2.84) and decisive evidence for including the main effect of congruency (BF(incl) = $1.64 \times 10^{14}$), with participants responding faster on congruent trials (M = 629 ms, SD = 81 ms) than on incongruent trials (M = 665 ms, SD = 84 ms). There was also moderate evidence for including the memory × congruency interaction (BF(incl) = 7.45).

Fig 3 depicts changes in the congruency effect with varying memory loads or differences in the memory effect between congruent and incongruent trials. A set of follow-up Bayesian ANOVAs was conducted to unpack the interaction. For congruent trials, there was moderate evidence *against* a memory-load effect (BF$_{01}$ = 4.05). In contrast, incongruent trials

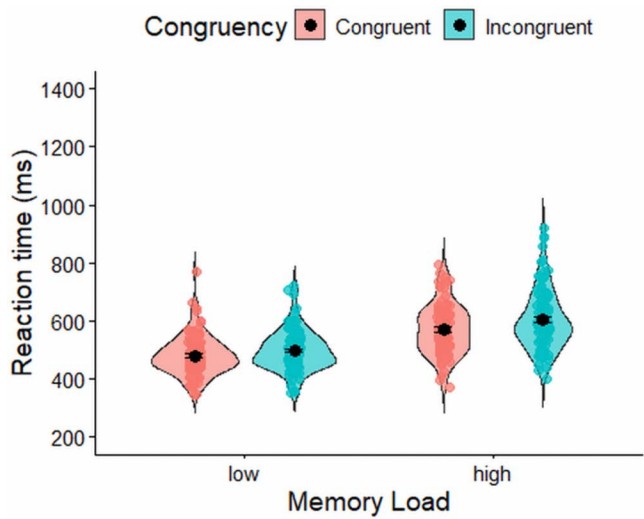

**Fig 2. Reaction time (ms) in the flanker task.** Error bars represent ±1 standard error of the mean (SEM).

**Table 2. Descriptive statistics of reaction time (RT; in milliseconds [ms]) in the spatial conflict task (N = 97).**

| Condition | Mean | Std. deviation | Minimum | Maximum |
|---|---|---|---|---|
| Low memory and congruent | 634 | 93 | 417 | 966 |
| Low memory and incongruent | 679 | 92 | 463 | 948 |
| High memory and congruent | 625 | 96 | 449 | 1015 |
| High memory and incongruent | 651 | 94 | 479 | 940 |

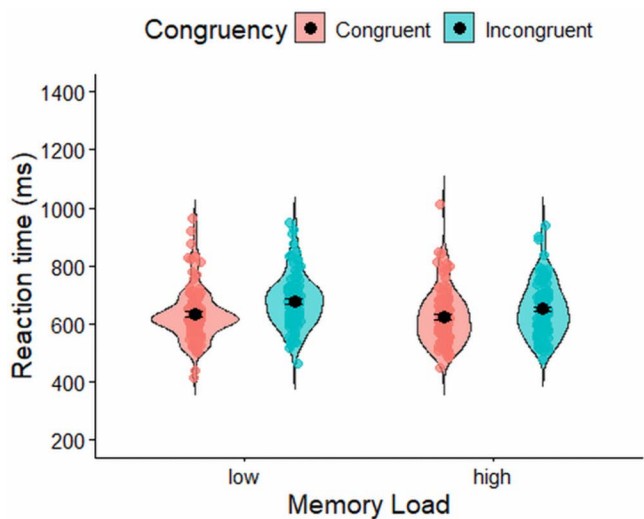

**Fig 3. Reaction time (ms) in the spatial conflict task.** Error bars represent ±1 standard error of the mean (SEM).

showed strong evidence supporting a memory-load effect ($BF_{10} = 52.45$). However, this effect was in the opposite direction to what was predicted, with participants responding faster under high memory load (M = 638, SD = 91) than under low memory load (M = 656, SD = 90).

## Correlations between the measures from the cognitive tasks and ASC traits

Bayesian correlation analysis examined the zero-order correlations between reaction times, accuracy, and inverse efficiency scores for congruent/incongruent trials within each low/high memory condition of each task, and the measure of ASC traits (including the AQ-S total score and each subscale score). The aim was to identify potential patterns of increasing correlations with increases in inhibitory and memory loads. In Study 1, the mean AQ-S total score was 62.23, with a standard deviation of 11.41 (detailed descriptive statistics for ASC traits, measured by AQ-S, are provided in the Supplementary Material in S3 Appendix). Bayesian correlation analyses consistently showed no strong evidence for a meaningful correlation between performance on the flanker and spatial conflict task conditions and AQ-S scores. Detailed results of the Bayesian correlations (including the posterior mean correlations, 95% credible intervals, and Bayes Factors) along with the correlation matrices showing associations between task measures of each task (including reaction times, accuracy, and inverse efficiency scores) and AQ-S scores, are provided in the Supplementary Material S3 Appendix (S3b).

## Partial correlations (Bayesian Linear regression)

As specified in our preregistration, a series of Bayesian linear regressions (equivalent to partial correlations) were conducted to examine whether the size of the congruency and memory effects in each task was associated with ASC traits. For each task, correlations were explored between incongruent trial performance across both low and high memory conditions, while controlling for congruent trial performance (thus providing an index of the size of the congruency effect) and ASC traits. Similarly, correlations were evaluated between high memory condition performance across both congruent and incongruent trials, while controlling for low memory condition performance (providing an index of the memory effect) and AQ-S scores.

For each regression, the Bayes Factor for inclusion (BF(incl)), the posterior mean of the coefficient, and the 95% credible interval are reported. BF(incl) reflects the evidence for including the predictor (AQ-S) compared with models that exclude it. Across all analyses, BF(incl) values were consistently below 1, posterior means were close to zero, and all 95% credible intervals included zero, indicating no evidence for meaningful associations between ASC traits and performance measures in either task. Table 3 summarises these results.

## Study 2

### Introduction

Study 2 was a complementary follow-up study aimed at developing specific aspects of Study 1. First, in Study 2, the direction of the central stimuli in the flanker task was adjusted by changing them to an upright position (see Fig 4). This adjustment was made because, despite observing an effect of distractor congruency in Study 1, the alignment of the central stimulus direction with the distractors meant that a congruency effect could potentially arise even if participants focused solely on the central stimulus and not the flankers. Second, and in contrast to Study 1 where the flanker task was the first task experienced by all participants, Study 2 divided participants into two groups: one in which the flanker task was the first task experienced and another in which the spatial conflict was the first task that participants received. This change was made because memory load effect were present (in the reaction time data) for the flanker task (the first task) but were not evident in the spatial conflict task (the second

**Table 3. Bayesian regression analyses (partial correlations) reporting associations between task performance effects and AQ-S scores.**

| Task | Partial correlation design | RT | Accuracy | Inverse efficiency scores |
|---|---|---|---|---|
| **Flanker task** | Incongruent trial performance and AQ-S (controlling for congruent) | BF(incl) = 0.025 | BF(incl) = 0.23 | BF(incl) = 0.056 |
| | | Mean = $-1.82 \times 10^{-6}$ | Mean = $-2.02 \times 10^{-4}$ | Mean = $5.26 \times 10^{-5}$ |
| | | 95% CI=[0.000, 0.000] | 95% CI=[−0.002, $2.23 \times 10^{-5}$] | 95% CI=[0.000, $2.16 \times 10^{-4}$] |
| | High memory performance and AQ-S (controlling for low memory) | BF(incl) = 0.37 | BF(incl) = 0.23 | BF(incl) = 0.192 |
| | | Mean = $9.49 \times 10^{-4}$ | Mean = $-9.27 \times 10^{-5}$ | Mean = $5.27 \times 10^{-4}$ |
| | | 95%CI=[$-8.42 \times 10^{-4}$, 0.007] | 95% CI=[−0.002, 0.002] | 95%CI=[$-3.85 \times 10^{-4}$, 0.007] |
| **Spatial Conflict task** | Incongruent trial performance and AQ-S (controlling for congruent) | BF(incl) = 0.191 | BF(incl) = 0.285 | BF(incl) = 0.323 |
| | | Mean = $2.035 \times 10^{-4}$ | Mean = $-3.744 \times 10^{-4}$ | Mean = $7.44 \times 10^{-4}$ |
| | | 95% CI=[$-3.061 \times 10^{-5}$, 0.002] | 95% CI=[−0.003, $6.402 \times 10^{-4}$] | 95% CI=[0.000, 0.006] |
| | High memory performance and AQ-S (controlling for low memory) | BF(incl) = 0.132 | BF(incl) = 0.155 | BF(incl) = 0.139 |
| | | Mean = $-1.159 \times 10^{-4}$ | Mean = $-5.94 \times 10^{-5}$ | Mean = $-2.54 \times 10^{-5}$ |
| | | 95%CI=[$-8.01 \times 10^{-4}$, 0.002] | 95%CI=[$-9.57 \times 10^{-4}$, $9.44 \times 10^{-4}$] | 95% CI=[−0.001, 0.002] |

Note. BF(incl) is the Bayes factor comparing models that include AQ-S with models that exclude it.

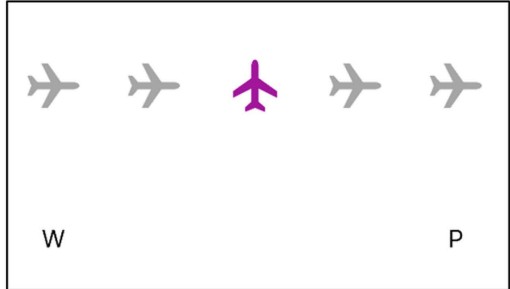
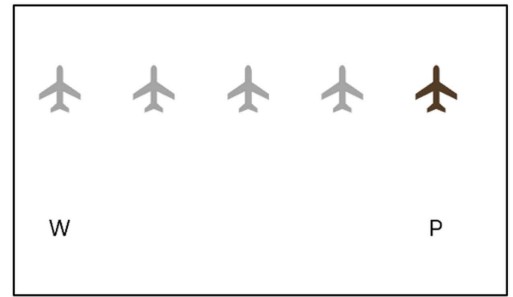

A sample trial in the Flanker task          A sample trial in the Spatial Conflict task

**Fig 4. Sample trials in cognitive tasks: Screenshots from the Gorilla program.**

task) and it is possible that the inconsistency could be due to the task order, as some colors from the flanker task were reused in the spatial conflict task, potentially leading to a carry-over effect. Third, to enhance the likelihood of observing a memory effect in both tasks, the quantity of stimuli requiring memorization in each condition was increased from four to six (the low memory condition involved six rules that could be grouped, while the high memory condition required memorizing six distinct, ungrouped rules; see Measures section for details). Finally, in Study 1, no significant correlation was found between autistic traits (AQ-S) and executive function task performance. Possible explanations for this lack of correlation include the absence of an inherent correlation, limitations in questionnaire sensitivity, and the potential influence of unmeasured ADHD traits. Given the high degree of overlap between ASC and ADHD [97], alongside the established association of ADHD with executive function difficulties [23,98,99], Study 2 expanded the trait assessments to include both ADHD and autistic traits. Additionally, to enhance the evaluation of autistic traits, the Comprehensive Autistic Trait Inventory (CATI, [100]) was employed instead of the AQ-S [89].

## Method

The Study 2 was preregistered on the Open Science Framework (OSF). Access to the preregistration documents for this study is available through the following link:
   Study 2: https://doi.org/10.17605/OSF.IO/HR7GY
This study received ethical approval from the School of Psychological Science Human Research Ethics Committee at the University of Bristol under the approval code 15238.

## Participants

The participants were 100 adults aged between 18 and 25 drawn from the general population, in accordance with the preregistered plan, ensuring a power of 94% for the detection of a correlation equal to or higher than 0.32. As in Study 1, applying the analysis-specific exclusion criteria led to different final sample sizes across analyses. The exact sample sizes for each analysis are reported in their respective sections.

   Participant recruitment for this study was conducted via the Prolific platform and participants were compensated for their involvement. The average reward offered per hour of participation was £13.19. Recruitment occurred between 27/06/2023 and 28/06/2023, with written informed consent obtained from all participants prior to the start of the study.

## Measures

   **Cognitive tasks (the executive function battery) and their conditions.** In Study 2 half of the participants began with the flanker task, and the remaining half started with the spatial conflict task. Both tasks were again administered through the Gorilla program. The design of Study 2 was the same as Study 1 except for the exceptions noted above.

The flanker task comprised a total of 308 trials, including 20 practice trials and 144 trials in each of low memory and high memory conditions and the spatial conflict task involved a total of 288 trials, including 24 practice trials and 132 trials in each of low memory and high memory conditions. The battery took approximately 30 minutes to complete. The manipulation of the executive function's components in Study 2 is described in detail below.

**Memory.**  To manipulate memory load, the complexity of the response rule that had to be maintained was varied in each condition. In the low memory conditions, participants were required to learn six rules, which had the potential to be grouped into three categories (e.g., W for light, mid, or dark blue airplanes; P for light, mid, or dark red airplanes). In the high memory load conditions, participants were instructed to remember six distinct rules. (e.g., W for brown, purple, or orange airplanes; P for yellow, green, or dark grey airplanes). The colours of the target airplanes for practice trials in both the flanker and spatial conflict tasks stayed the same (pink for W and gold for P). In the low memory trials of the flanker task, the chosen colours were light/mid/dark blue for W and light/mid/dark red for P and for the low memory trials of the spatial conflict task, the selected colours were light/mid/dark brown for W and light/mid/dark green for P. As in Study 1, colours were reused in high memory conditions due to limitations in the availability of distinct colours, maintaining their positions from low memory conditions. Consequently, for the high memory flanker task condition, brown, purple, and orange were assigned to W, and yellow, green, and dark grey were allocated to P. Within the high memory spatial conflict task condition, blue, purple, and orange were designated for W, and red, yellow, and dark grey were assigned to P.

**Inhibition.**  In Study 2, the manipulation of response inhibition in the spatial conflict task was identical to that in Study 1. All airplane stimuli were shown upright, with the coloured target appearing at either the far left or far right of the display. On congruent trials, the target appeared above the letter and response key it corresponded to (i.e., on the left for a W key press or on the right for a P key press). On incongruent trials, the target appeared on the side opposite its corresponding response (i.e., on the right for a W key press or on the left for a P key press).

In the flanker task, interference control was manipulated by varying the direction of the distractors around the upright central stimulus to be either incongruent or congruent with the correct response key. In congruent trials, the direction of the distractors matched the correct response key (e.g., distractors pointing left when the correct response was W). In incongruent trials, the distractors' direction conflicted with the correct response key (e.g., distractors pointing right when the correct response was W).

**Autistic traits measure.**  To assess autistic traits, in accordance with the preregistered plan, the CATI [100] was employed. The CATI is a relatively new test that measures a broad range of autistic traits, and it aims to correct some 'blind spots' that other common tests (e.g., AQ [90] and BAPQ [101]) have in the various autistic trait dimensions. The 42-item CATI comprises six subscales: Social Interactions, Communication, Social Camouflage, Repetitive Behaviours, Cognitive Rigidity, and Sensory Sensitivity. Items are rated on a 5-point Likert scale (1 = definitely disagree to 5 = definitely agree). In this study, the total CATI score served as the primary dependent variable, calculated by summing all 42 items (ranging from 42 to 210), with higher scores indicating greater levels of autistic traits. Although only the total score was preregistered for analyses, exploratory analyses were later conducted on all subscale scores in response to a reviewer request. The CATI has previously shown convergent validity at both the total-scale ($r \geq .79$) and subscale level ($r \geq .68$). The CATI has also shown superior internal reliability for total-scale scores ($\alpha = .95$) relative to the AQ ($\alpha = .90$) and BAPQ ($\alpha = .94$), consistently high reliability for subscales ($\alpha > .81$), greater predictive ability for classifying autism (Youden's Index = .62 vs.56 −.59) and has demonstrated measurement invariance for sex [100].

**ADHD traits measure.**  As preregistered, the Adult ADHD Self-Report Scale (ASRS, [102]) was utilized to measure ADHD traits. The ASRS is an 18-item self-report questionnaire designed to assess symptoms of ADHD in individuals aged 18 and above. This questionnaire divides into two subscales: a 9-item inattention subscale and a 9-item hyperactivity-impulsivity subscale. Items are rated on a 5-point Likert scale ranging from 0 (never) to 4 (very often). The total score, calculated by summing all 18 items (range: 0–72), was used as the primarily dependent variable in this study, with higher scores reflecting greater levels of ADHD traits. However, the hyperactivity and inattentive subscale scores were also

examined in exploratory analyses, as requested by a reviewer. The ASRS has high internal consistency (Cronbach's alpha = 0.88) and concurrent validity (r = 0.84) [103].

## Analysis plan

As in Study 1, Bayesian repeated-measures ANOVAs were run for each task to assess the effects of memory load, congruency, and their interaction. A set of alternative models was tested, ranging from the full model (including both factors and their interaction) to several restricted models that included each factor alone or both factors without the interaction. A null model with only subject (and random slopes) was also included. These models were compared using Bayes factors ($BF_{10}$) relative to the null model, and evidence for each effect was summarised using inclusion Bayes factors (BF(incl)).

Analyses were conducted separately for reaction time, accuracy, and inverse efficiency scores. Reaction time data were log-transformed to correct positive skew (a non-preregistered step), and these results are discussed in the main text. Accuracy outcomes are presented in Supplementary Material S6 Appendix. Inverse efficiency scores (also log-transformed), calculated to examine possible speed–accuracy trade-offs, are reported in Supplementary Material S7 Appendix. As in Study 1, this measure was added after preregistration based on reviewer feedback.

Subsequently, Bayesian correlation analyses examined the zero-order associations between performance measures (RT, accuracy, IE) and both ASC and ADHD traits. Follow-up analyses used Bayesian linear regressions to extract memory-load and inhibition-load effects, assessing their unique associations with ASC and ADHD traits via partial correlations. An additional set of non-preregistered analyses included age and gender as covariates (Supplementary Materials S12 Appendix for ASC, S14 Appendix for ADHD). Exploratory correlational analyses including all questionnaire subscales were also conducted (Supplementary Materials S11 Appendix for CATI, S13 Appendix for ASRS) in response to reviewer requests.

Additionally, Bayesian repeated-measures ANOVAs were conducted to examine whether task order influenced memory or congruency effects. Although some differences emerged, these effects were not consistent across tasks or performance measures (see Supplementary Material S8 Appendix).

As in Study 1, all analyses (Bayesian repeated-measures ANOVAs, correlations, and linear regressions) were conducted in JASP version 0.17.3 [92] using default priors and the "across matched models" option for Bayes Factor computation [93]. Model assumptions and diagnostics were examined, with no indications of serious violations.

## Outliers and exclusions

As preregistered, reaction-time outliers were identified using the MAD method [95], as in Study 1, and the corresponding trials were removed from all analyses. Additionally, Mahalanobis distance analyses were conducted on the four RT types (2 memory × 2 inhibition conditions) to identify participant-level outliers. Using a threshold of $p = .001$ ($\chi^2 > 18.47$, df = 4), two participants were excluded from the flanker task analysis and four from the spatial conflict task. As in Study 1, the Mahalanobis distance analyses were not preregistered and were conducted in response to reviewer feedback.

In addition, based on task performance, data from one participant were excluded from the flanker task dataset and two from the spatial conflict task dataset due to an average accuracy below 60% in the congruent trials of the low working memory load. As in Study 1, trials involving an immediate stimulus repetition were excluded from analysis.

Finally, as in Study 1, participants were asked for colour-blindness with the intention of excluding affected individuals, though this criterion was not preregistered. No participants in Study 2 were removed on this basis.

## Results

**Reaction time in the flanker task.** Table 4 displays the descriptive statistics for participants' reaction time in the flanker task.

**Table 4. Descriptive statistics of reaction time (in milliseconds) in the flanker task (N=97).**

| Condition | Mean | Std. deviation | Minimum | Maximum |
|---|---|---|---|---|
| **Low memory and congruent** | 538 | 114 | 362 | 925 |
| **Low memory and incongruent** | 551 | 123 | 364 | 948 |
| **High memory and incongruent** | 710 | 179 | 466 | 1374 |
| **High memory and congruent** | 699 | 171 | 449 | 1266 |

A Bayesian repeated-measures ANOVA on log-transformed reaction times showed that the best-fitting model included memory load, congruency, and their interaction ($BF_{10} = 3.92 \times 10^{22}$ relative to the null model). There was overwhelming evidence for including the main effect of memory load ($BF(incl) = 1.24 \times 10^{20}$), with participants responding more quickly in low-memory trials (M = 541 ms, SD = 124 ms) than in high-memory trials (M = 670 ms, SD = 183 ms). There was moderate evidence for excluding the main effect of congruency ($BF(excl) = 6.144$). In contrast, there was strong evidence for including the interaction between memory load and congruency ($BF(incl) = 1939.981$). Follow-up analyses decomposing this interaction revealed strong evidence for a congruency effect in the low-memory condition ($BF_{10} = 71.00$, relative to the null model), but only weak evidence for a congruency effect in the high-memory condition ($BF_{10} = 1.772$, relative to the null model). Fig 5 shows the reaction time distributions across low and high memory conditions and for congruent versus incongruent trials within each condition in the spatial conflict task.

**Reaction time in the spatial conflict task.** Table 5 displays the descriptive statistics for participants' reaction time in the spatial conflict task.

A Bayesian repeated-measures ANOVA on log-transformed reaction times in the spatial conflict task showed that the best-fitting model included memory load, congruency, and their interaction ($BF_{10} = 3.869 \times 10^{17}$ relative to the null model). There was very strong evidence for including the main effects of memory load ($BF(incl) = 6.026 \times 10^{10}$) and congruency ($BF(incl) = 1.259 \times 10^{5}$), with participants responding more quickly on trials with low memory load (M = 668 ms, SD = 122 ms) than on trials with high memory load (M = 766 ms, SD = 148 ms), and faster on congruent trials (M = 705 ms, SD = 124 ms) than on incongruent trials (M = 727 ms, SD = 123 ms). There was also strong evidence for including the interaction between memory load and congruency ($BF(incl) = 50.31$). Follow-up analyses decomposing this interaction showed very

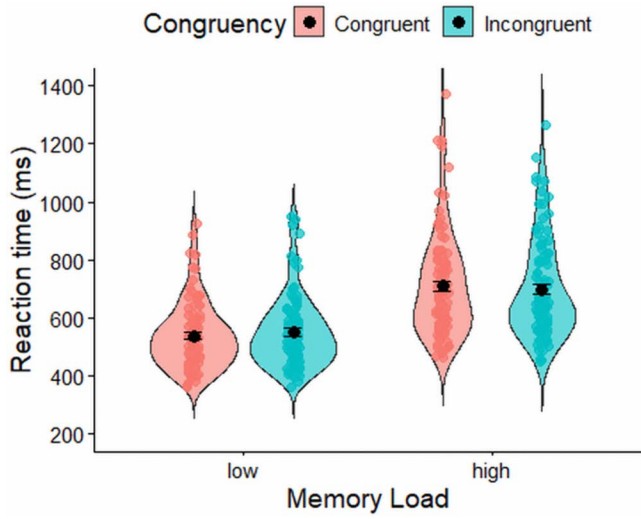

**Fig 5. Reaction time (ms) in the flanker task.** Error bars represent ±1 standard error of the mean (SEM).

**Table 5. Descriptive statistics of reaction time (in millisecond) in the spatial conflict task (N = 94).**

| Condition | Mean | Std. deviation | Minimum | Maximum |
|---|---|---|---|---|
| **Low memory and congruent** | 652 | 132 | 430 | 1061 |
| **Low memory and incongruent** | 684 | 118 | 482 | 976 |
| **High memory and congruent** | 759 | 147 | 532 | 1240 |
| **High memory and incongruent** | 772 | 156 | 532 | 1299 |

strong evidence for a memory effect in both congruent ($BF_{10} = 2.946 \times 10^{11}$) and incongruent ($BF_{10} = 7.916 \times 10^{7}$) trials. Additionally, there was very strong evidence for a congruency effect under low memory load ($BF_{10} = 1.306 \times 10^{6}$), but only weak evidence for a congruency effect under high memory load ($BF_{10} = 1.604$). Reaction time distributions for low and high memory conditions, and for congruent versus incongruent trials within each, are presented in Fig 6.

**Correlations between the measures from the cognitive tasks and ASC and ADHD traits.** A Bayesian correlation analysis showed strong evidence for a positive correlation between CATI and ASRS total scores, with a substantial correlation ($r = 0.49$), a 95% credible interval of [0.32, 0.62], and a very large Bayes factor ($BF_{10} = 62395$).

To investigate the correlation between the measures from the cognitive tasks and ASC and ADHD traits, a series of Bayesian correlation and partial correlation analyses were conducted. First, a series of Bayesian correlation analyses were conducted to examine the zero-order associations between ASC/ADHD traits and the performance measures (reaction time, accuracy, and inverse efficiency scores) across each task's conditions, including congruent and incongruent trials under both low and high memory load. These analyses investigated patterns of increasing correlations with an increase in inhibitory and memory load. In Study 2, the mean CATI score was 120.0 (SD = 30.58) and the mean ASRS score was 36.79 (SD = 14.10). Detailed descriptive statistics for ASC traits (measured by CATI) are presented in S9, and those for ADHD traits (measured by ASRS) are presented in S10. The Bayesian correlation analyses did not provide strong evidence for associations between ASC traits (either total scores or subscales) and any performance measures across any conditions of the flanker task. For the spatial conflict task, although negative correlations were observed between low-memory conditions (both congruent and incongruent) and the CATI Communication subscale, this evidence

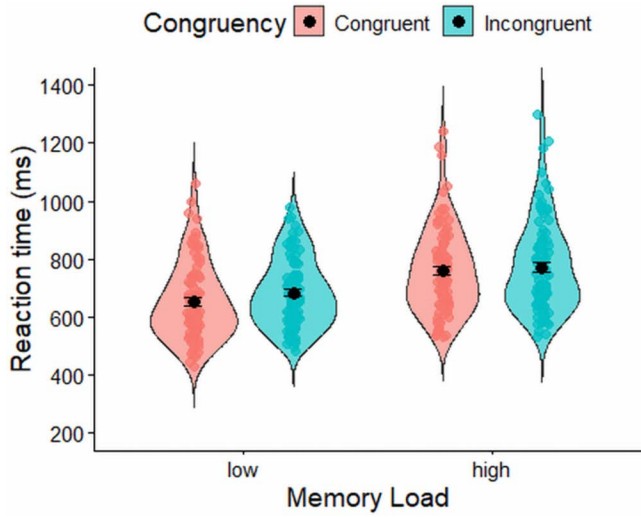

**Fig 6. Reaction time (ms) in the spatial conflict task.** Error bars represent ±1 standard error of the mean (SEM).

did not extend to the high-memory conditions. This pattern was contrary to what was expected of stronger negative correlations under higher task load (see Supplementary Material S9 Appendix).

For ADHD traits, there was no strong evidence for correlations between any ASRS subscale and any performance measure across task conditions (see Supplementary Material S10b Appendix). Taken together, there was no indication of an increase in the magnitude of correlations between either ASC or ADHD traits and task performance with increases in task load.

A series of partial correlations was then conducted to extract the effects of memory load and inhibition load and to examine their association with ASC/ADHD traits. Bayesian regression analyses were carried out for reaction time, accuracy as well as inverse efficiency scores in each task. Correlations were explored between incongruent trial performance across both low and high memory conditions, while controlling for congruent trial performance (producing an index of the congruency effect) and ASC/ADHD traits. Similarly, correlations were assessed between high memory condition performance across both congruent and incongruent trials, controlling for low memory condition performance (serving as an index of the memory effect) and ASC and ADHD traits. For each regression, BF(incl), the posterior coefficient mean, and the 95% credible interval are reported. BF(incl) assesses the evidence for including the CATI or ASRS scores as predictors. Across analyses, BF(incl) values were below 1 and the credible intervals included zero, indicating no reliable associations between these trait measures and task performance. Table 6 provides these findings.

## Discussion

This research consisted of two complementary studies each employing a novel test battery that incorporated two distinct tasks to simultaneously measure core components of executive function. Specifically, in both studies, a modified version of the Eriksen Flanker task was used to evaluate working memory and interference control simultaneously and a modified version of a 'Simon' spatial conflict task was employed to assess working memory and response inhibition concurrently. The 'modification' applied to both tasks was the introduction of two levels of memory load. The second study represented an improved version of the first, addressing certain issues identified in the initial study. These included: firstly, the orientation of the central stimuli in the flanker task. In Study 1, the central stimulus was oriented to the right or left, aligned with the flankers around it, whereas in Study 2, the central stimuli were positioned upright. A second difference involved the memory manipulation. In Study 1, participants learned four stimulus-response mapping rules. In Study 2, participants were presented with six stimulus-response mappings that could still be grouped into two rules in low memory conditions but that had to be maintained as six different and non-groupable rules in high memory conditions. A third difference was that participants in Study 2 were divided into two groups: half started with the flanker task, while the other half began with the spatial conflict task; in Study 1, all participants started with the flanker task. This change was based on the observation that memory load effects were present in the flanker task (reaction time data) but not in the spatial conflict task. It was hypothesized that the task order might have caused this inconsistency. A fourth difference was related to the participant pool: Study 2 recruited a more diverse sample through Prolific, while Study 1 included a more homogeneous group (all undergraduate psychology students). This broader recruitment in Study 2 likely contributed to the higher standard deviation in reaction times, but also increased the representativeness of the sample, which is a strength of the second study.

Overall, this research aimed to concurrently measure two core components of executive function – working memory and inhibitory control (specifically, either response inhibition or interference control in the corresponding task) – within a single paradigm. One objective was to determine if the main effects of these components can be extracted when assessed simultaneously and to investigate potential interactions between them. Additionally, this research explored the correlation between these executive function components and autistic and/or ADHD traits in the general population. A comprehensive analysis of accuracy data on each task is provided in the Supplementary Materials. Accuracy data from the flanker task in Study 1 showed limited evidence of a memory effect, which appeared only in congruent trials, and also indicated that a congruency effect (reflecting difficulties in interference control) was present only in the low memory

**Table 6. The correlation between the measures from the flanker/spatial conflict task and CATI/ASRS.**

| Task | Trial type | CATI | ASRS |
|------|-----------|------|------|
| **Flanker task** | Incongruent trial performance (within both low and high memory conditions) and controlling for congruent trial performance | **RT:**<br>BF(incl) = 0.005<br>Mean = $3.573 \times 10^{-7}$<br>95% CI=[0.000,0.000] | **RT:**<br>BF(incl) = 0.016<br>Mean = $4.596 \times 10^{-6}$<br>95% CI=[0.000,0.000] |
| | | **Accuracy:**<br>BF(incl) = 0.009<br>Mean = $4.478 \times 10^{-7}$<br>95% CI=[0.000,0.000] | **Accuracy:**<br>BF(incl) = 0.051<br>Mean = $-9.341 \times 10^{-6}$<br>95% CI=[$-1.930 \times 10^{-4}$,0.000] |
| | | **IE:**<br>BF(incl) = 0.006<br>Mean = $-1.151 \times 10^{-7}$<br>95% CI=[0.000, 0.000] | **IE:**<br>BF(incl) = 0.046<br>Mean = $2.413 \times 10^{-5}$<br>95% CI=[$-1.845 \times 10^{-5}$,$1.879 \times 10^{-4}$] |
| | High memory condition performance (including both congruent and incongruent trials) and controlling for low memory condition performance | **RT:**<br>BF(incl) = 0.061<br>Mean = $1.575 \times 10^{-5}$<br>95% CI=[$-2.638 \times 10^{-4}$,$1.112 \times 10^{-4}$] | **RT:**<br>BF(incl) = 0.141<br>Mean = $7.384 \times 10^{-5}$<br>95% CI=[$-0.003$,0.004] |
| | | **Accuracy:**<br>BF(incl)=0.119<br>mean = $-2.524 \times 10^{-5}$<br>95%CI=[$-0.0006371$,0.0002177] | **Accuracy:**<br>BF(incl)=0.208<br>mean=$-9.191 \times 10^{-6}$<br>95%CI=[$-0.002$,0.000] |
| | | **IE:**<br>BF(incl) = 0.079<br>Mean = $-1.865 \times 10^{-5}$<br>95% CI=[0.000, $8.100 \times 10^{-4}$] | **IE:**<br>BF(incl) = 0.171<br>Mean = $1.523 \times 10^{-4}$<br>95% CI=[$-0.002$,0.007] |
| **Spatial conflict task** | Incongruent trial performance (within both low and high memory conditions) and controlling for congruent trial performance | **RT:**<br>BF(incl) = 0.042<br>Mean = $-6.487 \times 10^{-7}$<br>95% CI=[0.000, 0.000] | **RT:**<br>BF(incl) = 0.022<br>Mean = $-2.376 \times 10^{-6}$<br>95% CI=[0.000,0.000] |
| | | **Accuracy:**<br>BF(incl) = 0.028<br>Mean = $5.167 \times 10^{-7}$<br>95% CI=[0.000,0.000] | **Accuracy:**<br>BF(incl) = 0.077<br>Mean = $-2.019 \times 10^{-5}$<br>95% CI=[$-2.563 \times 10^{-4}$,0.000] |
| | | **IE:**<br>BF(incl) = 0.009<br>Mean = $-2.038 \times 10^{-8}$<br>95% CI=[0.000,0.000] | **IE:**<br>BF(incl) = 0.021<br>Mean = $-3.752 \times 10^{-6}$<br>95% CI=[0.000, 0.000] |
| | High memory condition performance (including both congruent and incongruent trials) and controlling for low memory condition performance | **RT:**<br>BF(incl) = 0.061<br>Mean = $-2.100 \times 10^{-5}$<br>95% CI=[0.000, 0.000] | **RT:**<br>BF(incl) = 0.102<br>Mean = $-4.898 \times 10^{-5}$<br>95% CI=[$-6.821 \times 10^{-4}$,$7.267 \times 10^{-4}$] |
| | | **Accuracy:**<br>BF(incl) = 0.152<br>Mean = $-4.364 \times 10^{-5}$<br>95% CI=[$-4.214 \times 10^{-4}$,$1.220 \times 10^{-5}$] | **Accuracy:**<br>BF(incl) = 0.157<br>Mean = $2.768 \times 10^{-5}$<br>95% CI=[$-5.676 \times 10^{-4}$, $9.120 \times 10^{-4}$] |
| | | **IE:**<br>BF(incl) = 0.076<br>Mean = $2.728 \times 10^{-5}$<br>95% CI=[$-2.784 \times 10^{-5}$,$8.870 \times 10^{-4}$] | **IE:**<br>BF(incl) = 0.138<br>Mean = $-1.189 \times 10^{-4}$<br>95% CI=[$-0.003$, 0.002] |

Note. BF(incl) is the Bayes factor comparing models that include CATI or ASRS with models that exclude it.

condition. In Study 2, a memory effect was observed in accuracy, but no evidence for a congruency effect emerged. Accuracy data from the spatial conflict task in Study 1 showed no evidence of a memory effect but did reveal a congruency effect (reflecting response inhibition demands). In Study 2, a memory effect was observed in accuracy only for congruent trials, along with a congruency effect. However, accuracy may not be the most informative dependent variable in relatively straightforward tasks, where cognitive demands are low. In such tasks, there are often very few errors, even though accuracy is frequently measured [104]. In both of the current studies, participants maintained a high average accuracy.

Reaction time is therefore often the more informative measure in these contexts, not least because the current tasks are primarily concerned with speed of response. For reaction time in the flanker task, Study 1 showed both memory and congruency effects (reflecting difficulties in interference control). In Study 2, the orientation of the central stimuli was changed to an upright position. Although the memory effect persisted, the congruency effect was observed only in the low memory condition. One possible explanation for the absence of a congruency effect in the high memory condition in Study 2 could be the increased number of items to memorize, from 4 in Study 1–6 in Study 2. This may have caused participants to prioritize memorizing the central stimuli rules, possibly leading to less attention being allocated to the direction of the flankers surrounding them [105,106]. In the spatial conflict task, reaction time data from Study 1 showed clear congruency effects, indicating response-inhibition demands. A memory-load effect was also observed on incongruent trials, but in the opposite direction to what had been expected. Part of the reason for this counter-intuitive finding may be related to the memory manipulation: the four stimuli were intended to be grouped into two pairs in the low-load condition and to be non-groupable in the high-load condition. However, in the high-memory-load condition, W was mapped to blue and purple and P to red and yellow, which may have been interpreted as cool versus warm colour categories. This unintended categorisation could have reduced the intended working-memory demands. Some of the changes introduced in Study 2 aimed to increase the likelihood of a more consistent memory load effect. In line with this, there was clear evidence supporting the main effect of memory load in Study 2. Additionally, there remained strong evidence for the congruency effect. It needs to be noted that the presence of a congruency effect across both memory load conditions in the spatial conflict task in study 2, but only in the low memory condition in the flanker task, can be understood in terms of the distinct inhibitory processes each task engages.

It is worth noting that the flanker task primarily taps interference control, which involves resisting distraction from irrelevant surrounding stimuli. Under high memory load, and consistent with Lavie's load theory, participants are likely to prioritize memorizing the central stimulus–response rules, allocating less attention to the direction of the surrounding flankers. This narrowing of attention leaves fewer resources available to process or suppress external distractors, resulting in a reduced congruency effect. In contrast, the spatial conflict task relies more on response inhibition, which requires suppressing a prepotent motor response when the target's spatial location conflicts with the correct response key. Because the target appears unpredictably on either side of the screen, participants must continuously monitor both the memory rules and the target's position, preventing them from focusing solely on memory information. Consequently, both memory and congruency effects were evident in the spatial conflict task, whereas in the flanker task, the congruency effect emerged only under low memory load. However, it should be mentioned that in the flanker task, if anything there was a slight reduction in Study 2 in the size of the congruency effect in the low memory condition (13 ms compared to 17 ms in Study 1). Although these effects are small, they indicate that a meaningful congruency effect can still be observed even when the central stimulus is presented in an upright position.

In addition to accuracy and reaction time, a third dependent measure was added in response to reviewer feedback to evaluate possible speed–accuracy trade-offs. Inverse efficiency (IE = RT/ proportion correct) was computed for each task in both studies. The IE results closely matched the patterns observed for reaction time and were not inconsistent with the accuracy findings. This convergence across measures indicates that participants did not trade speed for accuracy in any task of either study. Full analyses of IE are presented in Supplementary Material S2 Appendix (Study 1) and S7 Appendix (Study 2).

In summary, in Study 2, the modifications made to the tasks enabled the successful extraction of key executive function effects when performance was measured in terms of reaction times. Specifically, working memory and response inhibition effects were successfully extracted in the spatial conflict task, while the flanker task successfully captured working memory and interference control effects (though the latter was limited to the low memory condition). These outcomes demonstrate the ability of these novel tasks to capture core components of executive function—working memory and inhibitory control (including both types of response inhibition and interference control)—reflecting their unique design that systematically and orthogonally manipulates memory load and either type of inhibitory demand. However, the absence of formal validation of these modified tasks relative to other independent measures of working memory and aspects of inhibition is acknowledged as a potential limitation, underscoring the need for future research to further establish their construct validity. Furthermore, it is also acknowledged that there is still potential for improving the task design, particularly in the flanker task, to increase the likelihood of observing a congruency effect under high memory load conditions (possible strategies to achieve this will be discussed later).

In addition, the specific design of this research allows for the investigation of how core components of executive function—working memory and either type of inhibitory control (response inhibition or interference control)—interact and influence each other. This exploration was prompted by the varying perspectives within current theories of executive function. Some frameworks conceptualize executive function as a unified system in which core processes draw on shared resources [72], while others view it as a set of distinct, fractionable components (see [76]). According to resource-sharing accounts, working memory and inhibition depend on a common, limited-capacity attentional system [72] and when demands on both processes are high, they compete for these shared resources, potentially leading to over-additive (interactive) effects. Conversely, models that conceptualize executive function as composed of separate but related components [71] predict additive rather than interactive effects of working memory and inhibition. The pre-registered hypotheses of the current research predicted that working memory and different types of inhibitory control are independent components and do not exhibit significant over-additive interactions in adult participants (see [9]).

No meaningful interactions between memory load and interference control manipulations of the flanker task were found for reaction time in Study 1 or for accuracy in Study 2. However, meaningful interactions were observed in the accuracy data for Study 1 (where a memory effect appeared only in congruent trials) and in the reaction time data for Study 2 (where the congruency effect was only evident under low memory load).

The analyses of spatial conflict task performance produced no meaningful interactions between working memory and response inhibition for accuracy in Study 1. However, meaningful interactions were evident in the reaction time data for both studies (though in Study 1 the memory effect was reversed and only present in incongruent trials, and in Study 2 there was only weak evidence for a congruency effect in high memory load) and in the accuracy data for Study 2 (where the memory effect was limited to congruent trials).

IE results consistently mirrored the interaction patterns observed for RT in each task in each study.

Therefore, interactions between working memory and either interference control or response inhibition were not consistently observed. In addition, when these did occur, they were under-additive, and no over-additive interactions were observed. In other words, high demands on both working memory and either type of inhibition did not produce the additional, over-additive decline in performance that would be expected under a shared-resource account. Overall, and as hypothesized, the data from this research did not provide good evidence of a meaningful over-additive interaction between the core components of executive function (working memory and either interference control or response inhibition). Instead, the findings suggest that these core components operate relatively independently, aligning with models that emphasize the distinctiveness of different executive functions (e.g., [76]).

The research also aimed to explore potential correlations between task performance (measured by accuracy, reaction time, and inverse efficiency scores) and autistic traits within the general population. Study 1 employed the short form of the Autism Quotient (AQ-S) to assess autistic traits. However, there was no evidence of an increase in degree of

meaningful zero-order correlation between ASC traits in the general population and any of task performance as memory and inhibitory load increased across conditions (see Supplementary Material S3 Appendix). Our main analysis employed a series of Bayesian regression correlations (equivalent to partial correlations) to extract memory load and inhibition load effects, and to investigate their connections with ASC traits more directly. These revealed no evidence of a meaningful correlation between the size of any congruency or memory effects and ASC traits. Furthermore, although not preregistered, additional Bayesian regressions were conducted at reviewer request including age and gender as covariates. These likewise showed no meaningful associations after adjusting for these factors (see Supplementary Material S5 Appendix).

This might reflect the fact that there is genuinely no underlying correlation between these contracts. Alternatively, the AQ-S' potentially limited sensitivity might have impeded the detection of correlations. Other studies have shown the AQ-S questionnaire to have good validity and reliability (e.g., [107–109]) and reported its relative success in capturing core dimensions of individual differences in traits associated with ASC. Additionally, the reliability of the AQ-S questionnaire was assessed in this study and found to be satisfactory, with a total McDonald's $\omega$ coefficient of 0.855. Nevertheless, Study 2 explored this issue further by employing the Comprehensive Autistic Trait Inventory (CATI, [100]), which potentially covers a broader spectrum of traits across various dimensions.

Study 2 also examined ADHD traits using the Adult ADHD Self-Report Scale (ASRS, [102]), as prior research on executive function in ASC may have unintentionally captured variation in ADHD-related traits instead of, or in addition to, pure autistic traits. Given the cooccurrence of ASC and ADHD, and the known executive function issues associated with ADHD, differences in executive function findings among ASC individuals in previous research could be partly due to concurrent ADHD symptoms. However, the correlational findings from Study 2 were very similar to those of Study 1. Zero-order correlations showed no good evidence of an increasing correlation between task performance (higher reaction times and lower accuracy) and ASC/ADHD traits in the general population as memory and inhibitory load increased across conditions (see Supplementary Material S9 Appendix for ASC and S10 Appendix for ADHD). Bayesian regression analyses that extracted memory-load and inhibition-load effects likewise showed no meaningful associations. The same null pattern emerged in additional Bayesian regressions that included age and gender as covariates, conducted at the reviewers' request (see Supplementary S12 Appendix for ASC and S14 Appendix for ADHD).

This clear lack of a relationship between experimental indices of memory and inhibitory load and ASC traits could be driven by various factors. It seems unlikely that the CATI and ASRS questionnaires lack the sensitivity needed to properly identify traits associated with autism and ADHD within the general population. Prior research has established the reliability of both measures, and they were meaningfully correlated with each other in Study 2. This correlation indicates that these two questionnaires reliably capture shared traits and exhibit consistent patterns of measurement, as well as providing further support for the strong overlap in these two sets of traits [110–117]. Similarly, it seems unlikely that the tasks used in this research failed to adequately measure core executive function components. The modifications made in Study 2 allowed for the successful measurement of these effects in the reaction time data. More specifically, memory and response inhibition effects were successfully extracted in the spatial conflict task, and in the flanker task memory and congruency effects (although only for low memory trials in the latter case) were also obtained.

Instead, a possible explanation for the lack of correlation with trait questionnaire measures could be that the tasks used in this research tap into specific aspects of executive function that are distinct from the traits assessed by the AQ-S, CATI and ASRS questionnaires. More specifically, while these questionnaires focus on broad traits, it could well be that our experimental tasks demand narrower and more specialized executive function skills. In general, task-based and questionnaire-based measures often show low correlations, even when measured reliably, because they potentially assess different dimensions of control. For instance, task-based measures are objective, focusing on immediate performance under ideal conditions with clear goals and feedback. In contrast, questionnaires rely on self-awareness and reflect long-term behaviours in real-world, often emotional, contexts. These fundamental differences in what and how they

measure explain the low correlation between tasks and questionnaires (see [118]). Another possible explanation, noted by Hedge et al. [119], is that well-established experimental effects derived from cognitive tasks might, almost by definition, lack the degree of individual differences needed to allow for correlations with other factors. Such tasks are designed to produce consistent results across individuals, which in turn leads to low variation in performance. This lack of individual variation makes it challenging to identify meaningful correlations. Therefore, the tasks used in this research may not give rise to effects that correlate with ASC and ADHD traits because the (original) Eriksen Flanker task and the Simon spatial conflict task were designed to yield consistent experimental outcomes rather than to capture individual differences. A further explanation is that individuals with high ASC and ADHD traits (similar to high-functioning neurodivergent individuals) may develop neural compensatory mechanisms to support cognitive challenges (including executive functioning difficulties) by using alternative cognitive routes to achieve performance levels comparable to their neurotypical peers [120,121]. For instance, Rahko et al. [122] found that while autistic adolescents performed attention and working memory tasks comparably to neurotypical peers, their brain activation patterns differed, indicating reliance on compensatory neural mechanisms. Similar patterns of neural compensation are also seen in ADHD (e.g., [123]). However, as task complexity increases, such compensatory strategies tend to become less effective, highlighting their limitations under higher task demands [122].

To address potential methodological limitations, several strategic approaches could be considered for future research. First, there is the potential to further improve our task design, particularly in the case of the flanker task. An effective approach could involve adjusting the colour contrast between the central target and the surrounding flankers. In the present design, where the central airplane was coloured and the flankers were grey, participants may primarily focus on the stimulus-response rules associated with the coloured central target and overlook the grey flankers. This tendency might be even higher under the high memory load condition, where the participant's limited cognitive capacity is directed towards maintaining a higher number of rules in memory. By reducing the contrast and introducing colour variation to both the central target and the flankers, participants could be prompted to concurrently divide their attention across both the memory and congruency aspects of the task.

Second, while this study adopted a transdiagnostic, dimensional approach by examining ASD and ADHD traits within the general population, future research could strengthen this by also including individuals with formal diagnoses to increase the likelihood of detecting correlations between these traits and task performance. Although this leans toward a more traditional, categorical approach, it could increase trait variability and improve the chances of identifying meaningful associations. Furthermore, it should be emphasized that although the dimensional approach taken in this study avoids binary groupings and aligns with current trends in neurodevelopmental research, the absence of any direct recruitment of participants with a clinical diagnosis necessarily limits the generalizability of the findings to clinically diagnosed neurodivergent populations.

Finally, the study did not include direct measures of potentially relevant factors such as sensory sensitivities, attention, emotional regulation, depression, anxiety and other cognitive domains which may influence executive function performance. It is important to note that a key design feature of our two tasks is that the 'low memory low inhibition' trials serve as a way of controlling for a range of factors that might affect general task performance, as the calculation of memory and inhibition load effects are made relative to this 'control' condition. Nevertheless, future research would benefit from incorporating other related measures to further take into account the potentially confounding effects of these factors on executive function outcomes.

## Conclusion

In summary, this research consisted of two complementary studies that aimed to examine the main effects of memory and inhibitory loads when these were assessed concurrently in each of two novel tasks, to investigate potential over-additive interactions between these executive function components, and to explore the correlation between ASC/ADHD traits and task performance. Each

involved 100 participants drawn from the general population, with the second study building upon the first. In Study 1, the flanker task showed the expected effects of working memory and congruency, while in the spatial conflict task only the congruency effect was as predicted. In Study 2, main effects of both working memory and congruency were successfully extracted in both tasks for reaction time, although the congruency effect in the flanker task was only evident in the low memory condition. Hence, these tasks hold potential for future research aiming to concurrently measure core executive function components in a pure manner. Furthermore, as hypothesized, an over-additive interaction between working memory and congruency was not evident in either of the two studies. Future research can employ more refined working memory tasks to better investigate potential interactions.

Additionally, in both studies, Bayesian correlation analyses and Bayesian linear regression found evidence against any meaningful correlations between the size of the congruency effect or the size of the memory effect, for either task and either dependent variable (accuracy, RT and IE), and ASC or ADHD traits. This lack of association may be due to the different methods used to measure ASD and ADHD traits (through questionnaires) and executive function components (through cognitive tasks), which are likely to capture related but distinct aspects of cognition and behaviour. Future research could explore alternative methods for measuring ASC traits, such as more specialized questionnaires, behavioural observations, clinician assessments, or physiological measures to gain clearer insights into these relationships.

## Supporting information

**S1 Appendix. Accuracy in the flanker task and spatial conflict task in Study 1.**
(DOCX)

**S2 Appendix. Speed–accuracy trade-off in Study 1.**
(DOCX)

**S3 Appendix. Correlations between the measures from the cognitive tasks and ASC traits (Study 1).**
(DOCX)

**S4 Appendix. Partial correlations between cognitive task measures and ASC traits (Study 1).**
(DOCX)

**S5 Appendix. Partial correlations between cognitive task measures and ASC traits with age and gender as covariates (Study 1).**
(DOCX)

**S6 Appendix. Accuracy in the flanker task and spatial conflict task in Study 2.**
(DOCX)

**S7 Appendix. Speed–accuracy trade-off in Study 2.**
(DOCX)

**S8 Appendix. Investigating memory and congruency effects based on task order.**
(DOCX)

**S9 Appendix. Correlations between the measures from the cognitive tasks and ASC traits (Study 2).**
(DOCX)

**S10 Appendix. Correlations between the measures from the cognitive tasks and ADHD traits (Study 2).**
(DOCX)

**S11 Appendix. Partial correlations between cognitive task measures and ASC traits (Study 2).**
(DOCX)

**S12 Appendix. Partial correlations between cognitive task measures and ASC traits with age and gender as covariates (Study 2).**
(DOCX)

**S13 Appendix. Partial correlations between cognitive task measures and ADHD traits (Study 2).**
(DOCX)

**S14 Appendix. Partial correlations between cognitive task measures and ADHD traits with age and gender as covariates (Study 2).**
(DOCX)

## Author contributions

**Conceptualization:** Yasamin Rahmati, Christopher Jarrold.

**Data curation:** Yasamin Rahmati.

**Formal analysis:** Yasamin Rahmati.

**Methodology:** Yasamin Rahmati, Christopher Jarrold.

**Resources:** Yasamin Rahmati.

**Software:** Yasamin Rahmati.

**Supervision:** Christopher Jarrold.

**Writing – original draft:** Yasamin Rahmati.

**Writing – review & editing:** Christopher Jarrold.

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
