## [Decision Letter · Decision Letter 0]

13 Jan 2025

Dear Dr. RAHMATI,

Thank you for submitting your manuscript to PLOS ONE. After careful consideration, we feel that it has merit but does not fully meet PLOS ONE’s publication criteria as it currently stands. Therefore, we invite you to submit a revised version of the manuscript that addresses the points raised during the review process.

We look forward to receiving your revised manuscript.

Kind regards,

Thiago P. Fernandes, PhD

Academic Editor

PLOS ONE

Journal requirements: When submitting your revision, we need you to address these additional requirements. 1. Please ensure that your manuscript meets PLOS ONE's style requirements, including those for file naming. The PLOS ONE style templates can be found at https://journals.plos.org/plosone/s/file?id=wjVg/PLOSOne_formatting_sample_main_body.pdf and https://journals.plos.org/plosone/s/file?id=ba62/PLOSOne_formatting_sample_title_authors_affiliations.pdf. 2. Please include a caption for figure 2. 

Reviewers' comments:

Reviewer's Responses to Questions

**Comments to the Author**

1. Is the manuscript technically sound, and do the data support the conclusions?

Reviewer #1: No

Reviewer #2: Yes

2. Has the statistical analysis been performed appropriately and rigorously?

Reviewer #1: No

Reviewer #2: Yes

3. Have the authors made all data underlying the findings in their manuscript fully available?

Reviewer #1: Yes

Reviewer #2: Yes

4. Is the manuscript presented in an intelligible fashion and written in standard English?

Reviewer #1: Yes

Reviewer #2: Yes

Reviewer #1: Is the manuscript technically sound, and do the data support the conclusions?

While the manuscript explores an important and timely topic—the interplay between executive functions (EF) and traits of Autism Spectrum Condition (ASC) and ADHD—it is compromised by several significant methodological, theoretical, and analytical limitations. These issues, outlined in detail below, raise substantial concerns about the robustness of the study's conclusions.

Critical Analysis of the Introduction and Justification

Oversimplification of Neurodiverse Conditions: The introduction treats neurodevelopmental conditions as homogeneous, failing to account for individual variability, such as the presence of compensatory mechanisms in high-functioning neurodivergent individuals or those with dual exceptionality. This omission leads to an overly reductive portrayal of EF deficits in these populations.

Ambiguity in Deficits Across EF Subcomponents: While the manuscript highlights EF deficits in ASC, ADHD, and Tourette’s Syndrome, it fails to specify which EF subcomponents (e.g., working memory, inhibitory control, cognitive flexibility) are affected. This lack of specificity weakens the scientific rigor of the argument.

Misrepresentation of EF in ASC Populations: The claim that all individuals with ASC exhibit EF deficits ignores evidence of preserved or enhanced EF in certain contexts. It also neglects the role of environmental and compensatory factors. Moreover, the absence of a robust control group limits the validity of these assertions.

Inconsistent Framing of Autistic Traits: By broadening its scope to include subclinical autistic traits, the manuscript does not clarify whether these traits directly influence EF or reflect normal cognitive variability. Without robust controls, this approach risks conflating variability in the general population with EF deficits specific to ASC.

Inadequate Consideration of Co-occurring Conditions: While discussing ASC-ADHD overlap, the manuscript neglects other relevant conditions, such as anxiety, depression, or sensory processing disorders, that may confound EF measures. Furthermore, equating genetic overlap with functional equivalence oversimplifies the complexity of shared EF deficits.

Methodological Concerns

Task Randomization and Order: Study 1 employed a fixed task order (Flanker first, followed by Spatial Conflict), potentially introducing fatigue or practice effects. Study 2 counterbalanced task order to address this, but the inconsistency undermines direct comparisons between studies.

Sample Representativeness: The sample comprises only neurotypical participants, limiting generalizability to neurodivergent populations. Furthermore, the absence of detailed demographic data (e.g., age, education, cultural background) limits interpretation of how individual variability may have influenced results.

Working Memory Manipulation: The increased stimulus mapping rules in Study 2 ostensibly heightened working memory demands, but without direct verification (e.g., recall testing), it remains unclear whether participants internalized these rules. This undermines the validity of the manipulation.

Task Modifications and Validation: The Flanker and Spatial Conflict tasks were modified to include memory components. However, no prior validation studies were presented, raising concerns about whether these tasks adequately measure the intended constructs (e.g., working memory-inhibitory control interaction).

Outcome Metrics: While reaction time (RT) and accuracy are standard metrics, the study does not address potential trade-offs between these measures (e.g., participants sacrificing speed for accuracy). This omission could distort interpretations of EF performance.

Statistical Analysis

High Variability in Results: The reported standard deviations (e.g., SD = 140ms in high-memory/incongruent conditions) are unusually high relative to the means, suggesting substantial uncontrolled variability that weakens the robustness of inferential statistics.

Lack of Control Variables: The study fails to include potential confounding variables such as IQ, baseline working memory capacity, or sensory sensitivities as covariates. These omissions make it difficult to attribute observed effects solely to the experimental manipulations.

Ambiguity in Bayesian Analysis: Although Bayesian ANOVA was employed, the manuscript does not address whether assumptions of data homogeneity and independence were satisfied. The lack of sensitivity analyses further undermines confidence in the reported Bayes factors.

Interpretation of Results

Non-Significant Interaction Effects: The absence of "over-additive" interactions between working memory and inhibitory control should have prompted a deeper exploration of task sensitivity limitations rather than being presented as an expected outcome. Additionally, alternative explanations, such as cognitive compensatory mechanisms, are not addressed.

Contradiction of the Transdiagnostic Model: Despite endorsing a transdiagnostic approach, the study relies heavily on categorical distinctions (e.g., ASC traits versus ADHD traits), undermining its theoretical coherence.

Limited Relevance to Neurodivergent Populations: As the findings are based exclusively on neurotypical participants, they do not directly inform our understanding of EF in neurodivergent populations. This translational gap is not adequately addressed in the discussion.

Reproducibility and Data Availability

Transparency: The authors provide comprehensive datasets, analysis scripts, and task instructions, demonstrating strong adherence to PLOS ONE’s open data policies.

Validation Deficiency: However, the absence of validation studies for task modifications (e.g., Flanker and Spatial Conflict adaptations) limits the interpretability and reproducibility of the results.

Recommendations for Improvement

Expand Participant Diversity: Future studies should include neurodivergent individuals alongside neurotypical participants to better capture EF variability across diagnostic boundaries.

Validate Task Modifications: Conduct validation studies to confirm that the modified Flanker and Spatial Conflict tasks reliably measure the intended constructs.

Improve Measurement Methods: Incorporate physiological or observational measures to complement self-reported data and reduce reliance on potentially biased questionnaires (e.g., AQ-S, CATI).

Control for Confounding Variables: Include measures of attention, sensory processing, emotional regulation, and other cognitive domains as covariates to refine interpretations of EF performance.

Strengthen Theoretical Framing: Develop hypotheses focusing on specific EF subcomponents and their differential impact across ASC, ADHD, and overlapping traits. Align these hypotheses with experimental design and analyses.

Overall Decision

The manuscript addresses an important topic and provides open access to its data and materials. However, significant methodological inconsistencies, lack of task validation, and limited translational applicability necessitate substantial revisions to meet PLOS ONE’s rigorous standards.

Recommendation: Major Revision Required

Reviewer #2: The manuscript is well-written, with a methodology aligned with the study's objective and a comprehensive literature review in the introduction that is skillfully integrated into the conclusions. The authors present two robust studies in the methodology section, offering a solid methodological design to test the proposed hypotheses. Statistical analyses are appropriate, and various supplementary materials are provided, allowing readers easy access—except for the OSF platform material associated with Study 2, which was inaccessible to me.

The writing is clear and intelligible, with well-structured content and grammatically correct English, free of typographical errors.

Overall, the manuscript provides valuable insights into the tripartite model of executive functions, emphasizing the working memory and inhibitory control components. It explores how these elements contribute to understanding neurodiverse conditions such as autism and attention-deficit/hyperactivity disorder (ADHD). The authors conclude that executive function (EF) tools differ fundamentally from those assessing ADHD and autism traits, which they consider the primary explanation for the observed lack of a relationship between traits and EF.

Comments on Study 1:

In lines 212–216, the instructions appear contradictory (or at least unclear) when compared to lines 261–262. It is advisable to clarify this section to prevent potential confusion for readers.

Regarding the instrument used to measure autistic traits (AQS), in lines 273–275, the authors describe the domains assessed by this tool and a factor derived from the mentioned dimensions. However, they later refer to the total score of the instrument without prior explanation or clarification of what high or low scores signify. To improve reader comprehension, it is recommended to clarify how the total score is derived and what it represents.

Similarly, in line 349, where the mean total score of the AQS is reported, the meaning of this value remains unclear. The methods section should include a clear explanation of how the total score is calculated and its interpretation.

Comments on Statistical Analysis:

The abbreviation "BF" is not defined anywhere in the methods section. It is recommended to specify what this abbreviation stands for when it first appears.

Comments on Study 2:

In the introduction section, line 388, the authors mention that the memory load conditions were increased from four to six items. However, it is unclear at this point what this entails or how the task with six items works. Since this aspect is described in the measures section, it would be helpful to clarify this earlier in the text and reference the section where the information is located.

**Do you want your identity to be public for this peer review?** For information about this choice, including consent withdrawal, please see our Privacy Policy

Reviewer #1: **Yes: ** Gabriel Boer Grigoletti Lima

Reviewer #2: **Yes: ** María Luisa García-Gomar

---

## [Author Response · Author response to Decision Letter 1]

11 Jun 2025

Reviewer 1

Reviewer 1’s comments have been interpreted and categorised into four main areas: Introduction, Methodological, Statistical Analysis, and Interpretation of Results. Each of these areas is summarises and addressed below.

• Introduction

Reviewer 1 raised several points regarding the Introduction. We have grouped these comments into two main categories, and they have been addressed throughout the revised Introduction.

1. Clarifying the Neurodiversity Approach and Addressing the Oversimplification and Overgeneralization of Executive Function in Neurodivergent Populations

a) Oversimplification of Neurodiverse Conditions: The introduction treats neurodevelopmental conditions as homogeneous, failing to account for individual variability, such as the presence of compensatory mechanisms in high-functioning neurodivergent individuals or those with dual exceptionality. This omission leads to an overly reductive portrayal of EF deficits in these populations.

b) Misrepresentation of EF in ASC Populations: The claim that all individuals with ASC exhibit EF deficits ignores evidence of preserved or enhanced EF in certain contexts. It also neglects the role of environmental and compensatory factors. Moreover, the absence of a robust control group limits the validity of these assertions.

c) Inadequate Consideration of Co-occurring Conditions: While discussing ASC-ADHD overlap, the manuscript neglects other relevant conditions, such as anxiety, depression, or sensory processing disorders, that may confound EF measures. Furthermore, equating genetic overlap with functional equivalence oversimplifies the complexity of shared EF deficits.

d) Inconsistent Framing of Autistic Traits: By broadening its scope to include subclinical autistic traits, the manuscript does not clarify whether these traits directly influence EF or reflect normal cognitive variability. Without robust controls, this approach risks conflating variability in the general population with EF deficits specific to ASC.

This has been addressed by better explaining the adopted neurodiversity perspective in the introduction. The revised text highlights individual differences in EF, acknowledges the presence of compensatory mechanisms, and references evidence of preserved EF in some autistic individuals. Contextual and environmental factors are also considered.

Regarding the lack of a control group, we now explain more clearly the benefits of taking a non-categorical approach to examining neurodiversity within the general population instead of making comparisons with a control group. We also note here how this approach aligns well with the sense of the reviewer’s Points A to C above). We trust that this justifies our approach sufficiently, but we have also highlighted (p. 38-39) the benefits of future work including both individuals with and without formal diagnoses and measuring potential confounding factors to statistically control for these.

Regarding the consideration of co-occurring conditions (Point C), there is a related danger that a strong focus on as anxiety and depression would push things back towards a more traditional diagnostic framing. While we fully appreciate that this is not the reviewer’s intention, this would again be in contrast to the dimensional, neurodiversity-informed perspective underpinning the current work. The revised text therefore clarifies our position by presenting co-occurring features as part of a broader, integrated neurodiversity framework rather than as separate categorical diagnoses. Our suggestion of controlling for individual variation on continuous measures of anxiety and depression in future work (p. 39) also directly addresses this concern.

The revised text also clarifies that subclinical traits are considered along a continuum within the general population, and that EF variability is interpreted within this broader neurodiversity framework, rather than being attributed solely to clinical thresholds (point D).

2. Lack of Specificity in EF Subcomponents

a) The manuscript does not specify which executive function subcomponents are affected across conditions.

The revised introduction now outlines the EF subcomponents most commonly impacted within the neurodiverse groups that form the focus of this paper.

• Methodological Concerns

a) Task Randomization and Order: Study 1 employed a fixed task order (Flanker first, followed by Spatial Conflict), potentially introducing fatigue or practice effects. Study 2 counterbalanced task order to address this, but the inconsistency undermines direct comparisons between studies.

The two studies were designed to be complementary rather than directly comparable. However, we did analyse potential task order effects in Study 2 (see supplementary material S6). While some differences emerged, the effects were not consistent across tasks or measures (reaction time vs. accuracy), suggesting that task order did not systematically bias the results.

b) Sample Representativeness: The sample comprises only neurotypical participants, limiting generalizability to neurodivergent populations. Furthermore, the absence of detailed demographic data (e.g., age, education, cultural background) limits interpretation of how individual variability may have influenced results.

In the revised Discussion, we emphasise that the sample consisted of participants who were not selected on the basis of any diagnosis and that we therefore examined correlations in the general population between ASD/ADHD traits and performance on our cognitive tasks. While we highlight the benefits of this approach, we do now explicitly note that this will limit the generalisability of the findings to clinically diagnosed neurodivergent populations (p. 34).

Regarding demographics, age data were collected for both studies. In Study 1, we also recorded educational background, as participants were undergraduate psychology students from the University of Bristol, and this information is now added to the Participant section of Study 1 (p. 11).

c) Working Memory Manipulation: The increased stimulus mapping rules in Study 2 ostensibly heightened working memory demands, but without direct verification (e.g., recall testing), it remains unclear whether participants internalized these rules. This undermines the validity of the manipulation.

While the novel design of the task for concurrently measuring working memory and inhibition did not include an explicit recall test, the structure of the task required participants to memorize the stimulus–response rules in order to respond correctly. Participants completed guided practice trials that provided accuracy feedback and repeated incorrect responses to support memorization before the main trials. Moreover, a pre-registered exclusion criterion (minimum 60% accuracy on low memory/congruent trials) ensured that only participants who had successfully memorized the rules were included in the analysis. The fact that nearly all participants met this threshold, alongside clear and meaningful reaction time differences observed between low and high memory load conditions in both tasks (particularly in Study 2, where the numbers of rules which needed to be memorized was increased), provides strong evidence that participants were indeed processing and applying the rules as intended (Strong evidence for memory effects in reaction times were observed in both tasks across both studies, though in Study 1, the effect in the spatial task was limited to incongruent trials, while in Study 2 it was more consistent).

However, we acknowledge that the task has not been formally validated against existing and accepted working memory measures (without reiterating our concerns over the validity of these ‘accepted’ measures), and this has now been stated as a limitation in the revised manuscript (p. 34) that future work can address.

d) Task Modifications and Validation: The Flanker and Spatial Conflict tasks were modified to include memory components. However, no prior validation studies were presented, raising concerns about whether these tasks adequately measure the intended constructs (e.g., working memory-inhibitory control interaction).

The tasks used in this study were based on well-established paradigms. The Spatial Conflict task was adapted from the classic Simon task, and the Flanker task was based on the Eriksen Flanker paradigm (both commonly used to assess interference control and response inhibition). Our versions simply incorporated an additional memory load component to explore the interaction between working memory and inhibitory control, following the approach used in Jarrold et al. (2023).

Although the tasks were modified, they consistently produced the expected effects: Both memory and congruency effects were observed in the Spatial Conflict task across studies, although the memory effect in Study 1 was limited to incongruent trials. Similarly, in the Flanker task, both effects were present, though in Study 2 the congruency effect emerged only under the low memory condition. The presence of these clear and robust effects, particularly in reaction time, provides strong evidence that the tasks effectively captured the intended constructs, including working memory demands, interference control, and response inhibition.

However, this discussion has now been expanded to explicitly acknowledge the lack of formal validation of the modified tasks against other more traditional measures as a limitation, and the need for future work to further establish their construct validity (p. 34).

e) Outcome Metrics: While reaction time (RT) and accuracy are standard metrics, the study does not address potential trade-offs between these measures (e.g., participants sacrificing speed for accuracy). This omission could distort interpretations of EF performance.

Thank you for this helpful observation. We agree that potential speed–accuracy trade-offs are important to consider when interpreting executive function performance. Although the pattern of results in the accuracy analyses did not contradict those in the RT analyses, we have conducted additional analyses to assess possible trade-off effects. These are now presented in Supplementary Materials S2 and S5. In brief, the results showed no evidence of a speed–accuracy trade-off in any task or condition.

• Statistical Analysis

a) High Variability in Results: The reported standard deviations (e.g., SD = 140ms in high-memory/incongruent conditions) are unusually high relative to the means, suggesting substantial uncontrolled variability that weakens the robustness of inferential statistics.

Thank you for this thoughtful comment. We would like to clarify that the RT data were already cleaned at the trial level using a robust and pre-registered trimming method. Specifically, we applied Median Absolute Deviation (MAD) trimming separately for each participant, within each task, memory condition, and trial type (congruent/incongruent).

However, in direct response to this comment, we further conducted a Mahalanobis distance analysis separately for each task within each study, based on the four RT trial types (2 memory × 2 inhibition). Participants whose values exceeded the critical threshold for p = 0.001 (χ² > 18.47, df = 4) were excluded. All related analyses have been updated accordingly after removing these cases within each task and each study.

Furthermore, we would like to note that standard deviations in the range of 130–140 ms are typical in cognitive tasks involving interference, particularly in incongruent conditions. For example, Medrano (2024), in ‘Flanker Task Performance in Young and Older Adults: A Behavioral and ERP Study’, and Duthoo & Notebaert (2012), in ‘Conflict Adaptation: It Is Not What You Expect’, report RT variability of similar magnitude. These findings reinforce our view the observed standard deviations in our study likely reflect genuine cognitive processing demands rather than uncontrolled noise.

Finally, we would like to emphasise that, even if the reported standard deviations appear relatively large, this can only limit our power to detect effects. Nevertheless, the main effects of interest were consistently observed across both studies, suggesting that the tasks remained sufficiently sensitive to capture the intended constructs. The variability in Study 2 likely reflects the more diverse and representative sample recruited through Prolific, as compared to the more homogeneous university sample in Study 1. This increased variability is not necessarily indicative of noise or error, but rather a reflection of natural variation in a broader population. This point has now been clarified in the revised discussion (p. 32).

b) Lack of Control Variables: The study fails to include potential confounding variables such as IQ, baseline working memory capacity, or sensory sensitivities as covariates. These omissions make it difficult to attribute observed effects solely to the experimental manipulations.

We acknowledge the value of including potential confounding variables. Indeed, in our design, the low-memory and congruent condition served as a within-subject baseline across all participants to compare performance in the other conditions. Furthermore, by comparing performance relative to this baseline, we are (although indirectly) controlling for stable individual differences, including general cognitive ability such as IQ. Since each participant acts as their own control, this approach reduces the influence of between-subject variability on the observed effects (a point we now make on p. 39).

However, given that we did not directly measure sensory sensitivities or other potentially confounding variables such as attention or emotional regulation, we have readily acknowledged this as a limitation in both the Discussion (p. 40) and Conclusion (p. 41) sections of the revised manuscript.

c) Ambiguity in Bayesian Analysis: Although Bayesian ANOVA was employed, the manuscript does not address whether assumptions of data homogeneity and independence were satisfied. The lack of sensitivity analyses further undermines confidence in the reported Bayes factors.

While the assumptions for Bayesian repeated measures ANOVA are somewhat less strict than those required for classical frequentist ANOVA, we confirm that the relevant assumptions, particularly the independence of observations, were met and have now been clearly stated in the revised manuscript (p. 17 and 26). Additionally, for repeated measures Bayesian ANOVA, the assumption of homogeneity of variance is replaced by the assumption of sphericity. In our design, with two within-subject’s factors each having only two levels, the assumption of sphericity is automatically satisfied and therefore not a concern.

• Interpretation of Results

a) Non-Significant Interaction Effects: The absence of "over-additive" interactions between working memory and inhibitory control should have prompted a deeper exploration of task sensitivity limitations rather than being presented as an expected outcome. Additionally, alternative explanations, such as cognitive compensatory mechanisms, are not addressed.

We acknowledge the potential for further improvements in task design, which has now been further clarified in the Discussion. However, particularly in Study 2, we observed robust main effects of our manipulations (albeit with the congruency effect being evident only under low memory load) indicating that the tasks were sensitive and the manipulations worked. The absence of an over-additive interaction therefore does not imply a lack of task sensitivity; rather, it suggests that working memory and response inhibition may operate in an under-additive or independent fashion under the current task demands.

Second, the use of Bayesian analysis in this study allows for a more rigorous interpretation of the absence of interaction than would be possible with traditional frequentist approaches. Rather than simply failing to detect an effect, the Bayes factors provide evidence against the presence/supporting the absence of an (over-additive) interaction.

Finally, this study was conducted in a general population sample and investigated ASC/ADHD traits dimensionally, in line wit

---

## [Decision Letter · Decision Letter 1]

17 Oct 2025

Dear Dr. RAHMATI,

Thank you for submitting your manuscript to PLOS ONE. After careful consideration, we feel that it has merit but does not fully meet PLOS ONE’s publication criteria as it currently stands. Therefore, we invite you to submit a revised version of the manuscript that addresses the points raised during the review process.

Please respond to all comments and highlight them in the revised ms.

We look forward to receiving your revised manuscript.

Kind regards,

Thiago P. Fernandes, PhD

Academic Editor

PLOS ONE

Journal Requirements:

Reviewers' comments:

Reviewer's Responses to Questions

**Comments to the Author**

Reviewer #3: All comments have been addressed

Reviewer #4: (No Response)

2. Is the manuscript technically sound, and do the data support the conclusions?

Reviewer #3: Yes

Reviewer #4: Partly

3. Has the statistical analysis been performed appropriately and rigorously?

Reviewer #3: Yes

Reviewer #4: Yes

4. Have the authors made all data underlying the findings in their manuscript fully available?

Reviewer #3: Yes

Reviewer #4: Yes

5. Is the manuscript presented in an intelligible fashion and written in standard English?

Reviewer #3: Yes

Reviewer #4: Yes

Reviewer #3: Dear Authors, Thank you very much for your contribution. After reading your manuscript, I found the content of the study very substantial and exciting. I look forward to the publication of your article, and wish you the very best!

Reviewer #4: Rahmati and Jarrold (2025) has two primary goals: (1) understanding the structure of different aspects of executive function and (2) investigating the relationship between executive function performance and psychiatric traits (specifically, ASD and ADHD traits). To accomplish this, the authors modify two standard executive function tasks (the Flanker task and a Spatial Conflict task), which typically measure two aspects of response inhibition, to parametrically modulate working memory load in addition to the inhibitory control demands. After categorizing the effects of the different facets of executive function (working memory and inhibitory control), the authors use a Bayesian correlation and partial correlation approach to relate individual differences in executive function performance to variation in psychiatric traits.

Although I commend the authors on a clever method to parametrically modulate different aspects of executive function and relate those to psychiatric traits measured in a transdiagnostic, continuous fashion, there were aspects of the framing and the analyses that made the results difficult to understand and interpret considering the broader literature.

Introduction:

Although the authors have attempted to incorporate additional literature in response to reviewers, the flow of the introduction is jumpy and difficult to follow – it jumps back and forth between discussing executive function in general to how executive function is impaired in psychiatric populations. It may make sense to streamline the introduction – introduce the concept of executive function and its importance, particularly for psychiatric populations, then go into why it is difficult to measure (both from the psychiatric perspective and from the measurement perspective) and finally address how the current study fills those gaps.

One thing that may help is being more structured with the various terminology – for instance, in the first sentence of the manuscript, the authors equate executive function to cognitive control. Later in in introduction in the discussion of the Norman and Shallice (1986) model, executive function and cognitive control seem to be described as two different constructs. Similarly, the authors could be more clear in their definition of working memory and inhibitory control earlier in the introduction, as these are their primary operationalizations of executive function (they are referenced in lines 58-59 before they are defined later in the paragraph around lines 63-66).

I also noted throughout the introduction and the discussion section that the authors reference a variety of models for the structure of working memory and inhibitory control (the Norman and Shallice model and the Baddeley model in the introduction, and the Roberts and Pennington model in the discussion) – it would be helpful to have these all presented in the same section, as this is a question the present work attempts to adjudicate.

In regards to the literature on ASD and ADHD, I commend the authors for adding in additional context as requested by previous reviewers. That said, the discussion of compensatory mechanisms may be more appropriate for the discussion section, especially when trying to explain the lack of a correlation between performance and psychiatric traits.

Although the authors do provide some description of their hypotheses, the term “over-additive” is not one that I’ve come across and would benefit from some additional clarification. It would be incredibly helpful if the authors included a more tangible hypothesis in terms of what changes one would expect in the differences in reaction time or accuracy across memory/inhibition conditions.

Some additional places where I did not understand what the authors were trying to say:

Line 62 – I’m not quite clear on what a “supportive mechanism” is

Line 88 – I did not know what the “general level of large, broad studies” – does this mean larger populations of individuals with ASD? Studies that include typically developing individuals? Wider age ranges?

Line 160 – it may be more clear to specify what kinds of sub-clinical traits can be present in non-formally diagnosed individuals, rather than simply stating “autistic traits,” especially given that autism is such a heterogenous disorder.

Line 176 – individual differences and phenotypic overlap does not necessarily impact EF research as a whole (as written), but studying EF in psychiatric populations

Task Design

I think it’s important to note that in Study 1, the high memory load condition could also be simplified to 2 rules – warm colors (yellow/red) vs cool colors (blue/purple).

What was the inter-trial interval? Did the task move between trials before a response was made? This is relevant because some of the responses were over a second long – if the response was longer than the ITI, it would make sense to trim/exclude.

Small comments:

- Were participants screened for color-blindness?

- It would be helpful to briefly reiterate how response inhibition was manipulated in the spatial conflict task in Study 2 (rather than just saying it was the same as Study 1).

Statistical Analyses

For the most part, I thought that the analyses presented in this manuscript were appropriate. That said, there are numerous places where additional information would provide clarity to the manuscript.

The authors seem to reference to testing different models in the results section but do not provide any reference as to what models were tested (what were the full and restricted models?) and how they were compared. This should be explicitly outlined.

Second, the data, as presented in tables and line charts, obscures the underlying distribution of data and the individual variability of the performance and the psychiatric traits. It would be helpful to have the accuracy/RT data presented as bar charts (rather than line plots), with dots reflecting mean performance/RT for each individual participant.

Moreover, the correlation plots of reaction time vs ASD/ADHD trait in Appendix S3 and S7 should be included as main figures – they allow the reader to inspect the distribution of performance and psychiatric traits. It would also be helpful for the interpretation of the relationship (or lack thereof) between psychiatric trait and executive function to note whether there is a threshold that implicates diagnosis (i.e. if the AQ-S has a range of scores from 28 to 112, would we expect someone who is diagnosed with ASD to score 50? 100?). Having this information helps contextualize the results and how they might generalize past the general population. It may also make the plots more intuitive to interpret if the trait score is on the x-axis and the performance measure is on the y-axis.

For the correlation analyses, the posterior mean correlation and 95% credible interval for the correlations should be provided, in addition to the Bayes Factor.

More detail should also be provided about the how the partial correlation analyses were conducted – were these true partial correlations, where the control variable (low working memory load or congruent trial condition) was regressed out of the outcome and then correlations were performed, or was a Bayesian linear regression performed (as was implied in a section heading from the Results section)? Either way, information about the posterior mean distribution and 95% credible interval should be provided in addition to the Bayes Factor (as mentioned for the zero-order correlations).

Finally, although the authors have added in an additional supplementary analyses regarding a speed accuracy trade-off, the analyses they completed do not quite hit the mark – one could presumably respond slower to ensure accurate performance on even the low memory load/congruent conditions. It would be more appropriate to calculate inverse efficiency (IE = RT/proportion correct responses) and use that as a dependent variable for analyses. This may be more sensitive to relationships with psychiatric trait, especially given potential for compensatory mechanisms in psychiatric populations.

One additional point – the authors mentioned that there were various subscales for the AQ-S and the ASRS – was the relationships between these sub-scores and the EF measures investigated at all? I could imagine that EF may be more relevant for certain diagnostic criteria for autism, or that the core diagnostic questions for the ASRS may be more relevant.

Particularly for the partial correlation analyses, were age and gender included as covariates of no interest? I could imagine that age may be particularly relevant, especially given literature that age is negatively correlated with working memory capacity, and developmental changes in EF in children.

A few final points of clarification:

- What priors were used for all Bayesian analyses?

- Were RT data log transformed prior to analysis (especially relevant given RT data are positively skewed) – this may also impact which priors should be used

- How many trials per participant were trimmed due to being classified as an outlier RT? Do results replicate if outlier trials are removed, versus being trimmed?

- For Table S7a, what does the column heading “Valid” mean?

- It should be mentioned that random slopes (and potentially subject specific random intercepts) were included in the JASP models by default (if this option was not turned off during analysis)

Discussion

After a cursory description of the inhibitory control constructs that the two tasks measure (Flanker task – interference control; spatial conflict – response inhibition), the differences are not discussed again and result are lumped into broader statements about inhibitory control. Especially given somewhat conflicting results across studies and tasks, and the framing in the introduction of inhibitory control being a multi-faceted construct, it may be worthwhile to discuss the differences in what the tasks are measuring and how that may have influenced results.

The discussion of Hedge’s paradox (lines 839-841) may be more appropriate placed around discussion of the sensitivity of the task.

The authors do a nice job explaining why the congruency effect only appeared in the low memory condition in the Flanker task in Study 2; it would be helpful to also have some speculation as to why there memory effect in the spatial conflict task in Study 1 was only present in incongruent trials (as this seems less intuitive as to why it might have occurred).

**Do you want your identity to be public for this peer review?** For information about this choice, including consent withdrawal, please see our Privacy Policy

Reviewer #3: No

Reviewer #4: No

---

## [Author Response · Author response to Decision Letter 2]

27 Nov 2025

Introduction

1. Although the authors have attempted to incorporate additional literature in response to reviewers, the flow of the introduction is jumpy and difficult to follow – it jumps back and forth between discussing executive function in general to how executive function is impaired in psychiatric populations. It may make sense to streamline the introduction – introduce the concept of executive function and its importance, particularly for psychiatric populations, then go into why it is difficult to measure (both from the psychiatric perspective and from the measurement perspective) and finally address how the current study fills those gaps.

Answer: The introduction has now been revised throughout to follow the recommended structure.

2. Similarly, the authors could be more clear in their definition of working memory and inhibitory control earlier in the introduction, as these are their primary operationalizations of executive function (they are referenced in lines 58-59 before they are defined later in the paragraph around lines 63-66).

Answer: This has now been moved earlier in the introduction, on page 3 (first page of the introduction), lines 44–51, to provide clearer and earlier definitions of working memory and inhibitory control.

3. I also noted throughout the introduction and the discussion section that the authors reference a variety of models for the structure of working memory and inhibitory control (the Norman and Shallice model and the Baddeley model in the introduction, and the Roberts and Pennington model in the discussion) – it would be helpful to have these all presented in the same section, as this is a question the present work attempts to adjudicate.

Answer: These models are now grouped together in the introduction (page 9, lines 164–197) and are briefly revisited in the relevant part of the discussion (page 42-43, lines 876–887).

4. the discussion of compensatory mechanisms may be more appropriate for the discussion section, especially when trying to explain the lack of a correlation between performance and psychiatric traits.

Answer: This discussion of compensatory mechanisms has now been moved to the discussion section (page 47, lines 984-993) to better explain the lack of correlation between performance and psychiatric traits.

5. Although the authors do provide some description of their hypotheses, the term “over-additive” is not one that I’ve come across and would benefit from some additional clarification.

Answer: Additional explanation of the term “over-additive” has been included on page 8 (lines 174–182).

6. It would be incredibly helpful if the authors included a more tangible hypothesis in terms of what changes one would expect in the differences in reaction time or accuracy across memory/inhibition conditions.

Answer: This has been addressed on page 10-11, lines 227–247, where we clearly outline the preregistered hypotheses, including the expected differences in reaction time, accuracy, and inverse efficiency scores across the memory and inhibition conditions.

Some additional places where I did not understand what the authors were trying to say:

1. Line 62 – I’m not quite clear on what a “supportive mechanism” is.

Answer: The sentence containing this term was removed as part of a larger edit to the introduction.

2. Line 88 – I did not know what the “general level of large, broad studies” – does this mean larger populations of individuals with ASD? Studies that include typically developing individuals? Wider age ranges?

Answer: This sentence was also deleted during the revision of the introduction.

3. Line 160 – it may be more clear to specify what kinds of sub-clinical traits can be present in non-formally diagnosed individuals, rather than simply stating “autistic traits,” especially given that autism is such a heterogenous disorder.

Answer: This has been clarified on page 6, lines 117–121.

4. Line 176 – individual differences and phenotypic overlap does not necessarily impact EF research as a whole (as written), but studying EF in psychiatric populations

Answer: This has been clarified on page 7, lines 139–140.

Method

Task design

1. It’s important to note that in Study 1, the high memory load condition could also be simplified to 2 rules – warm colours (yellow/red) vs cool colours (blue/purple).

Answer: We have now addressed this point in the Study 1 Methods section (p. 16, lines 361-364) by adding the following clarification:

“It is important to note that this colour set could potentially be categorized as warm versus cool colours, which may have reduced the intended working memory demand. This limitation was addressed in Study 2, considering the complementary nature of the two studies; see Study 2.”

2. What was the inter-trial interval? Did the task move between trials before a response was made? This is relevant because some of the responses were over a second long – if the response was longer than the ITI, it would make sense to trim/exclude.

Answer: There was no fixed inter-trial interval, as the task was fully response-initiated. The next trial began only after a response was made, and the task never advanced without participant input. Although participants were instructed to respond as quickly and accurately as possible, no external time limit was imposed. (Page 15, lines 346).

3. It would be helpful to briefly reiterate how response inhibition was manipulated in the spatial conflict task in Study 2 (rather than just saying it was the same as Study 1).

Answer: This has now been added and clearly explained (p. 29, line 594-599).

Inclusion/exclusion criteria

1. Were participants screened for color-blindness?

Answer: Yes. Before taking part in either study, all participants were asked to report whether they were colour-blind using the following screening question:

“Are you colour-blind? (Yes/No)”

This question was included with the intention of excluding colour-blind individuals, although no participants in either study reported being colour-blind. This information has now been added to the exclusion criteria sections of both studies (for Study 1: p.10, lines 441-443 and for Study 2: p.32, lines 667-679).

Statistical Analyses

1. The authors seem to reference to testing different models in the results section but do not provide any reference as to what models were tested (what were the full and restricted models?) and how they were compared. This should be explicitly outlined.

Answer: In the Bayesian repeated-measures ANOVA, we compared a set of models: a full model (with both main effects and their interaction), restricted models (lacking one or more effects), and a null model (including only subject effects and random slopes). We computed Bayes Factors to compare all models against the null model (BF₁₀). Furthermore, we derived inclusion Bayes Factors from this model set to quantify the evidence for each effect. The same model comparison approach was used for the Bayesian correlations and regressions. More detailed information is now provided in each relevant results sections.

2. Second, the data, as presented in tables and line charts, obscures the underlying distribution of data and the individual variability of the performance and the psychiatric traits. It would be helpful to have the accuracy/RT data presented as bar charts (rather than line plots), with dots reflecting mean performance/RT for each individual participant.

Answer: These figures (as well as the one for IE) have now been updated to violin plots with individual participant dots to better show the underlying distribution and variability.

3. Moreover, the correlation plots of reaction time vs ASD/ADHD trait in Appendix S3 and S7 should be included as main figures – they allow the reader to inspect the distribution of performance and psychiatric traits.

Answer: We appreciate the suggestion. However, as the analyses did not provide evidence for the proposed associations, we believe these plots are more appropriately kept in the Supplementary Materials rather than moved to the main text.

4. It would also be helpful for the interpretation of the relationship (or lack thereof) between psychiatric trait and executive function to note whether there is a threshold that implicates diagnosis (i.e. if the AQ-S has a range of scores from 28 to 112, would we expect someone who is diagnosed with ASD to score 50? 100?). Having this information helps contextualize the results and how they might generalize past the general population. It may also make the plots more intuitive to interpret if the trait score is on the x-axis and the performance measure is on the y-axis.

Answer: Although this study focuses on autistic and ADHD traits within the general population and adopts a transdiagnostic rather than categorical diagnostic approach, we now include the relevant cut-off information and mark these thresholds in the correlational graphs. We have also placed trait scores on the x-axis and performance measures on the y-axis (see S3 Appendix for Study 1, S9 Appendix for ASC in Study 2, and S10 Appendix for ADHD in Study 2).

5. For the correlation analyses, the posterior mean correlation and 95% credible interval for the correlations should be provided, in addition to the Bayes Factor.

Answer: We now report the posterior mean correlations and their 95% credible intervals alongside the Bayes Factors for each zero-order correlation (see S3 Appendix for Study 1, S9 Appendix for ASC in Study 2, and S10 Appendix for ADHD in Study 2).

6. More detail should also be provided about the how the partial correlation analyses were conducted – were these true partial correlations, where the control variable (low working memory load or congruent trial condition) was regressed out of the outcome and then correlations were performed, or was a Bayesian linear regression performed (as was implied in a section heading from the Results section)? Either way, information about the posterior mean distribution and 95% credible interval should be provided in addition to the Bayes Factor (as mentioned for the zero-order correlations).

Answer: They were conducted as Bayesian regression analyses (equivalent to partial correlations), and this is now stated clearly. We also report the posterior mean estimates and their 95% credible intervals alongside the Bayes Factors for these analyses (see S4 Appendix for Study 1 and S11 Appendix for ASC and S13 Appendix for ADHD in Study 2).

7. Finally, although the authors have added in an additional supplementary analyses regarding a speed accuracy trade-off, the analyses they completed do not quite hit the mark – one could presumably respond slower to ensure accurate performance on even the low memory load/congruent conditions. It would be more appropriate to calculate inverse efficiency (IE = RT/proportion correct responses) and use that as a dependent variable for analyses. This may be more sensitive to relationships with psychiatric trait, especially given potential for compensatory mechanisms in psychiatric populations.

Answer: We have now calculated inverse efficiency (IE = RT / proportion correct) and included it as an additional dependent variable in all relevant analyses. IE results (to assess potential accuracy–RT trade-offs) are provided in S2 Appendix for Study 1 and S7 Appendix for Study 2. Furthermore, IE is now used alongside RT and accuracy in all related correlation analyses, including both zero-order and partial correlations for each task in each study. The IE results indicated no speed–accuracy trade-off, and overall, IE almost mirrored the RT pattern.

8. One additional point – the authors mentioned that there were various subscales for the AQ-S and the ASRS – was the relationships between these sub-scores and the EF measures investigated at all? I could imagine that EF may be more relevant for certain diagnostic criteria for autism, or that the core diagnostic questions for the ASRS may be more relevant.

Answer: Subscale analyses are now reported in S3 Appendix for Study 1 and in S11 Appendix (CATI) and S13 Appendix (ASRS) for Study 2. These analyses revealed no evidence of meaningful correlations between the size of any congruency or memory effects and higher scores on any ASC or ADHD questionnaire subscales.

9. Particularly for the partial correlation analyses, were age and gender included as covariates of no interest? I could imagine that age may be particularly relevant, especially given literature that age is negatively correlated with working memory capacity, and developmental changes in EF in children.

Answer: Age and gender were not included as covariates in the preregistration, as all participants were adults within a narrow age range (18–25 years), minimising variability related to developmental or age-related differences in executive function. However, in response to the reviewer’s request, we conducted an additional set of non-preregistered analyses including age and gender as covariates (see S12 Appendix for ASC and S14 Appendix for ADHD). These Bayesian regressions showed the same null pattern as the original analyses.

Discussion

1. After a cursory description of the inhibitory control constructs that the two tasks measure (Flanker task – interference control; spatial conflict – response inhibition), the differences are not discussed again and result are lumped into broader statements about inhibitory control. Especially given somewhat conflicting results across studies and tasks, and the framing in the introduction of inhibitory control being a multi-faceted construct, it may be worthwhile to discuss the differences in what the tasks are measuring and how that may have influenced results.

Answer: This has been addressed throughout the discussion, where we now directly mention the specific type of inhibition being referred to in each relevant section. Furthermore, specifically, on page 41, lines 833-848, we note that the differences in the main effects observed between the two tasks may be related to the different types of inhibition they involve.

2. The discussion of Hedge’s paradox (lines 839-841) may be more appropriate placed around discussion of the sensitivity of the task.

Answer: We appreciate the suggestion and have given it careful thought, but we believe this point fits best in its current position in the discussion, where it directly supports the interpretation of the lack of correlation between the cognitive task and questionnaire-based measures of executive function.

3. The authors do a nice job explaining why the congruency effect only appeared in the low memory condition in the Flanker task in Study 2; it would be helpful to also have some speculation as to why there memory effect in the spatial conflict task in Study 1 was only present in incongruent trials (as this seems less intuitive as to why it might have occurred).

Answer: This memory effect occurred in the reverse direction (participants were faster in the high-load than the low-load condition, as now noted on p. 22, lines 481-483). One possible explanation for this counter-intuitive pattern relates to the memory-load manipulation. In the low-load condition, the four stimuli were designed to be grouped into two pairs, whereas in the high-load condition they were intended to be non-groupable. However, as the reviewer highlighted, in the high-load condition W was mapped to blue and purple and P to red and yellow, which may have encouraged categorical grouping into cool versus warm colours. This unintended categorisation could have reduced the expected working-memory demands of the high-load condition (now clarified in the manuscript, p. 40, line 823-829).

Throughout

1. One thing that may help is being more structured with the various terminology – for instance, in the first sentence of the manuscript, the authors equate executive function to cognitive control. Later in in introduction in the discussion of the Norman and Shallice (1986) model, executive function and cognitive control seem to be described as two different constructs.

Answer: Thank you for pointing that out. The term ‘executive function’ is now used consistently throughout the manuscript wherever that meaning was intended, ensuring clarity and alignment in terminology.

A

---

## [Decision Letter · Decision Letter 2]

14 Dec 2025

Concurrent measurement of working memory and inhibitory control and their correlations with autistic and ADHD traits in the general population

PONE-D-24-49592R2

Dear Dr. RAHMATI,

We’re pleased to inform you that your manuscript has been judged scientifically suitable for publication and will be formally accepted for publication once it meets all outstanding technical requirements.

Kind regards,

Thiago P. Fernandes, PhD

Academic Editor

PLOS One

Additional Editor Comments (optional):

Reviewers' comments:

Reviewer's Responses to Questions

**Comments to the Author**

Reviewer #4: All comments have been addressed

2. Is the manuscript technically sound, and do the data support the conclusions?

Reviewer #4: Yes

3. Has the statistical analysis been performed appropriately and rigorously?

Reviewer #4: Yes

4. Have the authors made all data underlying the findings in their manuscript fully available?

Reviewer #4: Yes

5. Is the manuscript presented in an intelligible fashion and written in standard English?

Reviewer #4: Yes

Reviewer #4: (No Response)

**Do you want your identity to be public for this peer review?** For information about this choice, including consent withdrawal, please see our Privacy Policy

Reviewer #4: No

---

## [Editor Report · Acceptance letter]

PONE-D-24-49592R2

PLOS One

Dear Dr. RAHMATI,

I'm pleased to inform you that your manuscript has been deemed suitable for publication in PLOS One. Congratulations! Your manuscript is now being handed over to our production team.

Kind regards,

on behalf of

Dr. Thiago P. Fernandes

Academic Editor

PLOS One